# LATENT REASONING WITH RECURRENT DEPTH FOR SEQUENTIAL RECOMMENDATION

## ABSTRACT

Sequential recommender systems play a pivotal role in modern applications by modeling user behavior sequences to predict their preferences. However, current approaches primarily adopt non-reasoning paradigms, which constrain their computational capacity and lead to suboptimal performance. To overcome these limitations, we propose LARES, an innovative and scalable **LA**tent **R**easoning framework for **S**equential Recommendation that unlocks deep thinking with a recurrent depth. Unlike conventional parameter scaling methods, LARES enhances the model's representational power by increasing the computational density of parameters through depth-recurrent latent reasoning. Its recurrent architecture allows flexible expansion of reasoning depth without extra parameters, thereby effectively capturing complex and evolving user interest patterns. To fully exploit the model's reasoning potential, we introduce a two-stage training strategy: (1) Self-supervised pre-training (SPT) with *trajectory-level alignment* and *step-level alignment*, where the model learns latent reasoning patterns tailored for sequential recommendation tasks without annotated data, and (2) Reinforcement post-training (RPT), which leverages reinforcement learning (RL) to encourage exploration of diverse reasoning paths and further refine its reasoning capabilities. Extensive experiments on real-world benchmarks demonstrate LARES's superiority. Notably, the framework exhibits seamless compatibility with existing advanced models, consistently improving their recommendation performance. Our code is available at `https://anonymous.4open.science/r/LARES-E458/`.

## 1 INTRODUCTION

In the era of information explosion, recommender systems have emerged as indispensable components across real-world applications ranging from e-commerce platforms to online streaming services (Singer et al., 2022; Zheng et al., 2024). Among them, current research has increasingly focused on the analysis of evolving user behaviors for capturing latent intentions and sequential patterns to predict future interactions, which is termed as sequential recommendation. Recent years have witnessed significant advancements in this area, with notable methods like SASRec (Kang & McAuley, 2018) and BERT4Rec (Sun et al., 2019), which adopt transformer-based architectures for improved user behavior modeling.

To cope with the intricate and dynamic nature of user behaviors, extensive research (Zhai et al., 2024; Xu et al., 2025a; Zhang et al., 2024) has been conducted to scale the computational capacity of sequential recommenders, thereby strengthening their representational power. Previous works (Zhang et al., 2024; Zhai et al., 2024; Yan et al., 2025) have primarily pursued this goal through parameter scaling. However, these efforts have failed to replicate the success achieved in large language models (LLMs). This discrepancy primarily stems from unique challenges in recommendation systems, including inherent data sparsity and quality limitations that impede effective model scaling (Zhang et al., 2024).

Recent advances in large reasoning models (DeepSeek-AI et al., 2025; Team et al., 2025; Chen et al., 2025b) have demonstrated that scaling test-time computation is another effective approach to increasing the utilization of computing power and can significantly enhance the reasoning capabilities of LLMs. This suggests a promising approach to improving model performance by increasing the computation density for each parameter. There are two primary paradigms for test-time scaling

in LLMs: one is explicit reasoning (Guo et al., 2024; Shao et al., 2024; Yang et al., 2024; 2025), where models verbalize intermediate reasoning steps (*i.e.,* chain-of-thoughts (CoTs)) by generating meaningful tokens before producing final answers, and the other is latent reasoning (Hao et al., 2024; Xu et al., 2025b; Geiping et al., 2025), where models perform multi-step implicit reasoning in the latent space without generating explicit reasoning tokens. While most LLMs adopt explicit CoT reasoning, this approach faces challenges in recommendation systems. Unlike LLMs operating in *dense* textual spaces, most sequential recommenders are confined to *sparse* item ID spaces. This fundamental difference makes it hard to define meaningful reasoning steps like CoTs and provide supervision signals for training (Lightman et al., 2024; Luo et al., 2024). Therefore, we adopt the latent reasoning approach to scaling sequential recommenders. Recent work like ReaRec (Tang et al., 2025) has demonstrated the potential of this paradigm through autoregressive generation of implicit reasoning tokens. Despite its effectiveness, this method remains computationally suboptimal as it only enriches *one* token per reasoning step.

To tackle this problem, we aim to fully unleash computation by leveraging *all* input tokens at each reasoning step. Our approach is inspired by the recently proposed depth-recurrent model architecture for latent reasoning (Geiping et al., 2025), which designs a recurrent block consisting of Transformer layers. It repeatedly applies the recurrent block to update the hidden states of all the tokens in the latent space for test-time compute scaling. For the scaling of sequential recommenders, since we also aim to unleash the computation of each item token in the latent space, it is feasible to develop a depth-recurrent architecture over existing recommendation models.

To this end, we propose **LARES**, a novel and scalable **LA**tent **RE**asoning approach for **S**equential recommendation that enables flexible test-time scaling by thinking in continuous latent spaces. Our approach adopts a recurrent architecture comprising two key components: a pre-block for initial processing and a core-block for iterative refinement. The core-block supports arbitrary iteration depth, allowing for dynamic computation scaling while improving computational density (*i.e.,* the amount of computation per parameter). To fully unlock the model's reasoning capabilities, we develop a two-stage training strategy, the self-supervised pre-training (SPT) and reinforcement post-training (RPT). During the SPT stage, we propose the trajectory-level alignment and step-level alignment objectives to equip the model with recommendation-oriented latent reasoning patterns. Specifically, in trajectory-level alignment, we want to achieve knowledge transfer between high-quality reasoning processes. In step-level alignment, we aim to improve the thinking coherence among different intermediate steps. During the RPT stage, we employ reinforcement learning to further refine the model's reasoning capabilities for recommendation.

In summary, our work makes the following main contributions:

• We propose **LARES**, a novel scalable latent reasoning approach for sequential recommendation that leverage all the input tokens to perform multi-step reasoning in latent space with arbitrary depth.

• We design a two-stage training strategy including SPT and RPT to fully unleash the model's reasoning capabilities. During SPT, we introduce trajectory-level and step-level alignment to improve reasoning coherence. For RPT, we leverage RL to further improve recommendation performance via task-aligned rewards.

• Extensive experiments on real-world benchmarks validate LARES's superiority, demonstrating its effectiveness and seamless compatibility with existing sequential recommenders.

## 2 METHODOLOGY

In this section, we present our scalable latent reasoning approach for sequential recommendation, named **LARES**, which is illustrated in Figure 1.

### 2.1 OVERVIEW

**Sequential Recommendation.** In recommender systems, there are a set of users $\mathcal{U}$ and a set of items $\mathcal{V}$. Let $M = |\mathcal{U}|$ and $N = |\mathcal{V}|$ denote the size of the user set and item set. The behavior record for each user $u \in \mathcal{U}$ is defined as an interaction sequence $S_u = [v_1, \ldots, v_n]$ at the time step $n$, where items are arranged in chronological order. As a typical sequential recommendation setting,

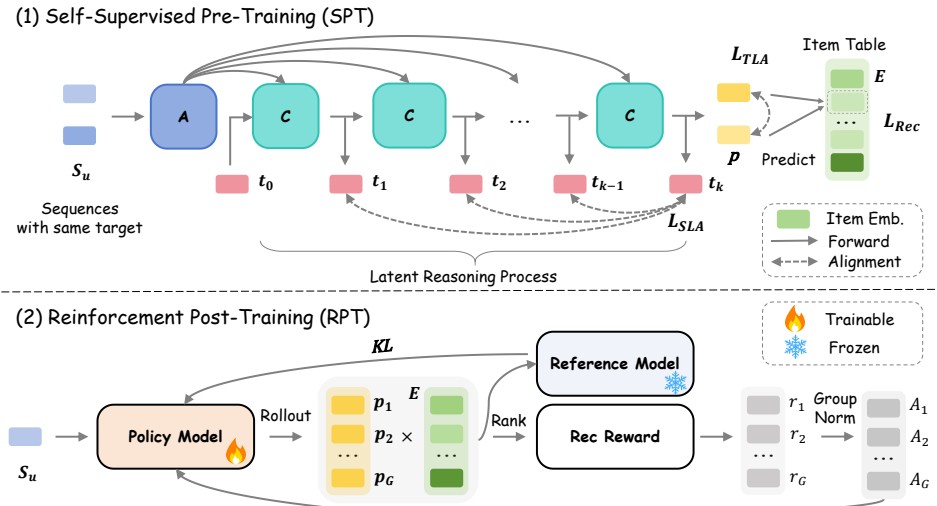

Figure 1: Overall framework of LARES. $\mathcal{A}$ and $\mathcal{C}$ denote the pre-block and core-block, repsectively. "TLA" and "SLA" represent the *Trajectory-Level Alignment* and *Step-Level Alignment*.

traditional sequential recommenders (Kang & McAuley, 2018; Zhou et al., 2020; Sun et al., 2019) encode the interaction sequence by direct inference to obtain sequential behavior representations. The recommender then predicts the next item the user is most likely to interact with based on the similarities between the encoded user representations and candidate item representations. The objective of next item prediction can be formally written as:

$$\max_{\Theta} P(v_{n+1}|S_u; \Theta), \tag{1}$$

where $\Theta$ denotes the parameters of the sequential recommender.

**Latent Reasoning For Sequential Recommendation.** Traditional sequential recommenders typically adopt a straightforward inference pattern to directly provide recommendations, which struggle with complex recommendation tasks. In contrast, recent studies (Tang et al., 2025) propose a latent reasoning sequential recommender which is capable of thinking in a continuous space before making recommendations. This multi-step thinking paradigm allows the model to iteratively refine its inference in latent space and obtain more accurate user interests, simulating human-like mental thinking processes when addressing challenging problems. Generally, the model progressively derives a series of intermediate thoughts, denoted as $T_u$ and makes final recommendations conditioned on these latent thoughts as well as the behavior sequence. Formally, the objective of latent reasoning sequential recommenders can be formulated as:

$$\max_{\Theta} P(v_{n+1}|S_u; \Theta) = P(v_{n+1}|T_u, S_u; \Theta) \cdot P(T_u|S_u; \Theta). \tag{2}$$

Existing work (Tang et al., 2025) models latent reasoning $P(T_u|S_u; \Theta)$ as an autoregressive generation process, where a new state is generated and appended to the input at each reasoning step, which can be represented as follows:

$$P(T_u|S_u; \Theta) = \prod_{i=1}^{k} P(t_i|S_u, t_{<i}; \Theta), \tag{3}$$

where $T_u = [t_1, \ldots, t_k]$, $k$ is the number of reasoning steps and $t_i$ and $t_{<i}$ denote the generated thoughts at the $i$-th step and preceding the $i$-th step, respectively.

**Scaling Latent Reasoning With Recurrent Depth.** To fully unleash the computing power of deep thinking in sequential recommendation, we propose LARES, a novel latent reasoning scaling paradigm with recurrent depth where *all* input tokens are refined at each reasoning step instead of

only generating *one* new token. The reasoning process of LARES is denoted as follows:

$$P(T_u|S_u;\Theta) = \prod_{i=1}^{k} P(S_u^i|S_u^{i-1}, S_u^0;\Theta), \tag{4}$$

where $T_u = [S_u^1, \ldots, S_u^k]$, $k$ is the number of reasoning steps (*i.e.,* recurrent depth), $S_u^i$ is the thought at the $i$-th step, and $S_u^0$ corresponds to the initial input. For notational simplicity, in the remainder of this paper, we will denote $S_u^i$ as $t_i$. The overall framework of LARES is depicted in Figure 1. It adopts a depth-recurrent design consisting of a pre-block for mapping initial features into latent space and a core-block iterable to arbitrary depths, enabling flexible test-time scaling without additional parameters. To facilitate effective latent reasoning with only the outcome label (*i.e.,* the next item), we propose a two-stage training approach for LARES: self-supervised pre-training for thinking adaptation and reinforcement post-training for thinking exploration. This approach draws inspiration from established practices in LLMs, where pre-training enables LLMs to acquire fundamental knowledge, while reinforcement learning incentivizes the reasoning capability for complex reasoning tasks (Jaech et al., 2024; DeepSeek-AI et al., 2025; Team et al., 2025; Chen et al., 2025b). Accordingly, our framework first employs self-supervised pre-training to equip LARES with the core capability of user interest modeling through iterative latent reasoning, then applies reinforcement post-training to further incentivize its reasoning capability by exploring diverse latent thought patterns.

## 2.2 Latent Reasoning with A Recurrent Architecture

Our proposed framework, LARES, is a depth-recurrent sequential model consisting of multiple Transformer layers, which can be adapted to other advanced sequential recommendation architectures, *e.g.,* FMLPRec (Zhou et al., 2022) and TedRec (Xu et al., 2024), as proved in the experiments in Section 3.4.1. In LARES, there are two blocks, *i.e.,* the pre-block $\mathcal{A}$ and the core-block $\mathcal{C}$. The pre-block first transforms the item embeddings into the latent space, and then the core-block performs a multi-step reasoning on the latent representations in a recurrent pattern. Finally, the last latent thought representation is employed for subsequent item prediction. The most significant design of LARES lies in the recurrence of the core block, which allows the model to dive into a deeper thinking of user behaviors without introducing extra parameter burdens. This design is inspired by the findings that the reasoning performance is up to a large effective depth but not necessarily many parameters (Saunshi et al., 2025; Geiping et al., 2025). The advantages of the recurrent architecture are twofold: on the one hand, it enhances the model's computational expressiveness with no extra parameter memory cost; on the other hand, it enables more flexible inference scaling by controlling the recurrent depth.

Suppose the historical behavior sequence of user $u$ is $S_u = [v_1, \ldots, v_n]$. Given the item embedding table $\boldsymbol{E} \in \mathbb{R}^{N \times d}$, the sequence $S_u$ is first transformed into item embeddings $\boldsymbol{E}_u = [\boldsymbol{e}_1; \ldots; \boldsymbol{e}_n]$ by table lookup, where $d$ is the embedding dimension and $[;]$ denotes the concatenation operation. First, $\boldsymbol{E}_u$ along with the position embeddings $\boldsymbol{E}_p \in \mathbb{R}^{n \times d}$ is fed into the pre-block $\mathcal{A}$ to produce the initial latent item representation $\boldsymbol{H} \in \mathbb{R}^{t \times d}$, which is written as: $\boldsymbol{H} = \mathcal{A}(\boldsymbol{E}_u + \boldsymbol{E}_p)$. Then, the recurrent core-block $\mathcal{C}$ performs iterative latent reasoning, taking $\boldsymbol{H}$ and the latent thought of the previous step $\boldsymbol{T}_{i-1}$ as inputs, formulated as:

$$\boldsymbol{T}_i = \mathcal{C}\left(\text{LN}\left(f\left(\boldsymbol{T}_{i-1}, \boldsymbol{H}\right)\right)\right), \text{ for } i \in \{1, \ldots, k\}, \tag{5}$$

where $\boldsymbol{T}_0 \sim \mathcal{N}(0, \sigma_1^2 I)$, $i$ and $k$ denote the $i$th step and the number of total reasoning steps, $f$ is the aggregation function *e.g.,* addition and concatenation, $\text{LN}[\cdot]$ denotes the layer normalization operation and $\sigma_1$ is the standard deviation of the normal distribution for initializing the random state $\boldsymbol{T}_0$. The input $\boldsymbol{H}$ to core-block $\mathcal{C}$ in every reasoning step functions as a residual connection to ensure stable gradient backpropagation. To allow flexible scaling of inference-time reasoning, during training, we sample the iteration count $k$ per training step from a *log-normal Poisson distribution*. For a target mean iteration count $\bar{k} + 1$ (where $\bar{k} \in \mathbb{N}$) and variance $\sigma_2$, the sampling procedure is defined as follows:

$$\xi \sim \mathcal{N}\left(\log(\bar{k}) - \frac{1}{2}\sigma_2^2, \sigma_2^2\right), \quad k \sim \mathcal{P}(e^\xi) + 1, \tag{6}$$

where $\mathcal{N}$ and $\mathcal{P}$ denote the normal and Poisson distributions, respectively. In our experiments, we set $\sigma_2 = 0.5$. This distribution predominantly samples values below $\bar{k}$, while retaining a heavy tail that

occasionally yields significantly higher iteration counts. To mitigate excessive memory consumption, we truncate $k$ such that $k = \min(k, 3\bar{k})$. In inference, the recurrent depth is set to $\bar{k} + 1$ for all test samples.

Finally, the final latent thought state corresponding to the last item $v_{n+1}$ is regarded as the final user preference representation $\boldsymbol{p} \in \mathbb{R}^d$ denoted as $\boldsymbol{p} = \boldsymbol{T}_k[-1]$. The output $\boldsymbol{p}$ is then used to compute the recommended probabilities of candidate items $\hat{y}_i$, and we adopt the widely used cross-entropy loss as the recommendation objective, which is formulated as:

$$\mathcal{L}_{\text{Rec}} = -\log \hat{y}_{n+1} = -\log \frac{\exp(\boldsymbol{p} \cdot \boldsymbol{e}_{n+1}^T)}{\sum_{i=1}^{N} \exp(\boldsymbol{p} \cdot \boldsymbol{e}_i^T)}, \tag{7}$$

where $\boldsymbol{e}_{n+1}$ denotes the target item embedding of $v_{n+1}$ that user $u$ will interact with at the time step $n + 1$.

### 2.3 SELF-SUPERVISED PRE-TRAINING FOR THINKING ADAPTATION

In the self-supervised pre-training (SPT) stage, LARES learns to perform recommendation-oriented latent reasoning for user interest modeling. However, relying solely on the recommendation objective $\mathcal{L}_{\text{Rec}}$ is not enough to ensure effective training because it cannot provide sufficient signals for the intermediate latent thinking process of LARES. To alleviate this problem, we propose two self-supervised optimization objectives, *i.e., trajectory-level alignment* and *step-level alignment*, to provide auxiliary supervision.

**Trajectory-Level Alignment.** To strengthen LARES's reasoning capability, we introduce trajectory-level alignment to leverage complementary strengths from different reasoning trajectories. We define different trajectories from two aspects: On the one hand, stochastic elements like random initialization and dropout naturally lead to varied reasoning paths across forward passes for the identical input sequences; On the other hand, the trajectory of different sequences sharing the same target item can also be regarded as positive views (Qiu et al., 2022). Specifically, we align the final outputs of independent reasoning trajectories between positive pairs. However, our experiments reveal that aligning two reasoning outcomes with different step lengths adversely impacts model performance. We posit that this occurs because the inconsistency in reasoning steps causes a misalignment between long-chain and short-chain reasoning. Specifically, forcing short-chain reasoning to capture the richer and more complex patterns in longer reasoning processes is inherently challenging. As a result, the model tends to degenerate toward short-chain reasoning, ultimately compromising its effectiveness. To address this, we ensure consistent reasoning steps within each positive pair. Formally, given two positive sequences $S_u$ and $\hat{S}_u$ and a shared reasoning step $k$, LARES produces final preference representations $\boldsymbol{p} = \boldsymbol{T}_k[-1]$ and $\hat{\boldsymbol{p}} = \hat{\boldsymbol{T}}_k[-1]$. We achieve the alignment between them based on the InfoNCE loss. The trajectory-level alignment objective is formulated as:

$$F(\boldsymbol{x}, \boldsymbol{y}^+, \mathcal{B}_{\boldsymbol{y}}) = -\log \frac{\exp(s(\boldsymbol{x}, \boldsymbol{y}^+)/\tau)}{\sum_{\boldsymbol{y} \in \mathcal{B}_y} \exp(s(\boldsymbol{x}, \boldsymbol{y})/\tau)}, \quad \mathcal{L}_{\text{TLA}} = \frac{1}{2}\left(F(\boldsymbol{p}, \hat{\boldsymbol{p}}, \mathcal{R}_{\hat{\boldsymbol{p}}}) + F(\hat{\boldsymbol{p}}, \boldsymbol{p}, \mathcal{R}_{\boldsymbol{p}})\right), \tag{8}$$

where $F(\cdot, \cdot, \cdot)$ denotes the InfoNCE loss function, $s(\cdot, \cdot)$ is a similarity metric (*e.g.,* cosine or dot product), $\mathcal{R}_{\boldsymbol{p}}$ and $\mathcal{R}_{\hat{\boldsymbol{p}}}$ represent sample sets containing both positive and negative instances, and $\tau$ is a temperature coefficient controlling the distribution sharpness.

**Step-Level Alignment.** Given a historical interaction sequence $S_u$ and a sampled iteration number $k$, LARES generates a sequence of latent thought representations $\boldsymbol{T}_u = [\boldsymbol{t}_1, \ldots, \boldsymbol{t}_k]$. Ideally, these latent thoughts are progressively refined to converge toward the true user preference distribution. While the model is expected to produce increasingly accurate latent representations as reasoning advances, intermediate states may occasionally diverge from the desired trajectory, leading to suboptimal or counterproductive reasoning steps. To address this issue, we introduce *step-level alignment* to enforce coherence between intermediate states and the final output. Specifically, we uniformly sample an intermediate step $b$ from $\{1, \ldots, k-1\}$ and optimize the alignment between $\boldsymbol{t}_b$ and $\boldsymbol{t}_k$ using an InfoNCE loss:

$$\mathcal{L}_{\text{SLA}} = \frac{1}{2}\left(F(\boldsymbol{t}_b, \boldsymbol{t}_k, \mathcal{B}_k) + F(\boldsymbol{t}_k, \boldsymbol{t}_b, \mathcal{B}_b)\right), \quad b \sim \text{Unif}\{1, 2, \ldots, k-1\}, \tag{9}$$

where Unif denotes the uniform distribution and $\mathcal{B}_i, i \in \{b, k\}$ contains the $i$-th step latent reasoning representations of all instances in the same batch.

The overall objective for self-supervised pre-training is written as:

$$\mathcal{L}_{\text{SPT}} = \mathcal{L}_{\text{Rec}} + \alpha \mathcal{L}_{\text{TLA}} + \gamma \mathcal{L}_{\text{SLA}}, \tag{10}$$

where $\alpha$ and $\gamma$ denote hyper-parameters to balance the weights among different objectives during optimizations, respectively.

## 2.4 Reinforcement Post-Training for Thinking Exploration

After the self-supervised pre-training stage, the model acquires latent reasoning patterns for sequential recommendation tasks. However, this stage suffers from limited thinking exploration due to the absence of supervisory signals that guide the model in distinguishing between "good" and "not good" reasoning steps. Consequently, the model's exploratory potential remains underutilized. To mitigate this limitation, we introduce a reinforcement learning-based post-training approach that enhances the model's reasoning capabilities through learning from experiences of high-quality reasoning trajectories. The subsequent sections introduce the reinforcement learning algorithm, reward design, and data selection startegy.

**Reinforcement Learning Algorithm.** To balance performance and computational cost, we employ the Group Relative Policy Optimization (GRPO) algorithm (Shao et al., 2024) during reinforcement post-training. For each input (*i.e.,* user interaction sequences $S_u$ in our case), GRPO samples a group of rollouts from the old policy $\pi_{\theta_{\text{old}}}$. The current policy $\pi_\theta$ (*i.e.,* the base model) is then updated by maximizing a reward function and regularized by the KL divergence from a reference policy $\pi_{\text{ref}}$ (*i.e.,* the initial pre-trained model). The objective of GRPO is formulated as:

$$\mathcal{J}_{\text{GRPO}}(\theta) = \mathbb{E}_{x \sim D, \{y_i\}_{i=1}^G \sim \pi_\theta} \left[ \frac{1}{G} \sum_{i=1}^G \left( \min \left( P \cdot A_i, C \cdot A_i \right) - \beta \, \mathbb{D}_{\text{KL}} \left( \pi_\theta || \pi_{\text{ref}} \right) \right) \right], \tag{11}$$

$$P = \frac{\pi_\theta(y_i|x)}{\pi_{\theta_{\text{old}}}(y_i|x)}, C = \text{clip} \left( \frac{\pi_\theta(y_i|x)}{\pi_{\theta_{\text{old}}}(y_i|x)}, 1 - \epsilon, 1 + \epsilon \right), \tag{12}$$

where $A_i$ is the advantage value, $x$ is the input, $y_i$ is the response generated by LLMs, and $\pi_\theta(y_i|x) = \prod_{j=1}^{|y_i|} \pi_\theta(y_{i,j}|x, y_{i,<j})$ is the generation probability of the response $y_i$ under policy $\pi$. However, directly applying Eq. equation 11 to our task is infeasible because it requires computing the joint probability $\pi(y_i|x)$ with discrete tokens, which are absent in our setting. To resolve this issue, we reformulate it as the joint probability of recommending the target item at each reasoning step, denoted as $\pi(y_i|x) = \pi(v_{n+1}|S_u) = \prod_{j=1}^k \pi(v_{n+1}|S_u, t_{i,j})$ where $v_{n+1}$ is the target item, $S_u$ denotes the input user sequence, $t_{i,j}$ represents the latent thought representation at $j$-th step of the $i$-th rollout and $k$ is the total number of reasoning steps.

**Reward Design.** To maintain strict alignment with recommendation objectives, we directly employ standard recommendation metrics (namely NDCG@$k$ and Recall@$k$) as reward signals. We take the recommendation results of the last reasoning step and the target label to calculate these metrics. Specifically, for $G$ rollout trajectories, we first take the last latent thought states corresponding to the last item in the input sequence as the final user representations $\{p_1, \ldots, p_G\}$ and then obtain the recommendation probability distribution for each rollout by computing the similarity between $p_i$ and the item embedding table $E$, denoted as $P_i = \text{softmax}(p_i \cdot E^T) \in \mathbb{R}^N, i \in \{1, \ldots, G\}$. Then we can obtain the item ranking list based on the probability distribution for every rollout, denoted as $L_i = \text{argsort } P_i$. The reward function is formally written as: $r_i = m(v_{n+1}, L_i), \quad m \in \{\text{NDCG@}k, \text{Recall@}k\}$. The advantage value $A_i$ is computed as the z-score normalized reward within each group: $A_i = \frac{r_i - \text{mean}(\{r_1, \ldots, r_G\})}{\text{std}(\{r_1, \ldots, r_G\})}$.

**Data Selection.** Recent work demonstrates that the difficulty of data is important to the effectiveness of RL (Team et al., 2025). Since labels in sequential recommendation are very sparse, it is important that the data have a balanced difficulty for a limited number of rollouts. Considering this, we propose a data selection strategy that filters out hard training samples. Specifically, we exclude instances where the pre-trained model fails to rank the target item within the top 100 positions across three independent inference trials.

Table 1: Performance comparison of different methods. The best and second-best results are indicated in bold and underlined font, respectively. "*" denotes that the improvements are statistically significant with $p < 0.01$ in a paired t-test setting.

| Dataset | Metric | GRU4Rec | BERT4Rec | SASRec | FMLP-Rec | CL4SRec | DuoRec | BSARec | ERL | PRL | PRL++ | LARES |
|---|---|---|---|---|---|---|---|---|---|---|---|---|
| Instrument | Recall@5 | 0.0318 | 0.0289 | 0.0346 | 0.0366 | 0.0354 | 0.0381 | 0.0363 | 0.0342 | 0.0345 | 0.0385 | **0.0411***  |
| | Recall@10 | 0.0514 | 0.0463 | 0.0536 | 0.0575 | 0.0552 | 0.0598 | 0.0564 | 0.0546 | 0.0551 | 0.0587 | **0.0636*** |
| | Recall@20 | 0.0774 | 0.0697 | 0.0798 | 0.0858 | 0.0831 | 0.0891 | 0.0841 | 0.0813 | 0.0834 | 0.0875 | **0.0934*** |
| | NDCG@5 | 0.0207 | 0.0182 | 0.0216 | 0.0233 | 0.0227 | 0.0244 | 0.0231 | 0.0216 | 0.0222 | 0.0245 | **0.0263*** |
| | NDCG@10 | 0.0271 | 0.0238 | 0.0277 | 0.0300 | 0.0291 | 0.0314 | 0.0295 | 0.0282 | 0.0288 | 0.0310 | **0.0336*** |
| | NDCG@20 | 0.0336 | 0.0297 | 0.0343 | 0.0372 | 0.0361 | 0.0388 | 0.0365 | 0.0349 | 0.0359 | 0.0382 | **0.0410*** |
| Scientific | Recall@5 | 0.0205 | 0.0183 | 0.0248 | 0.0250 | 0.0261 | 0.0280 | 0.0267 | 0.0245 | 0.0258 | 0.0279 | **0.0297*** |
| | Recall@10 | 0.0340 | 0.0310 | 0.0385 | 0.0404 | 0.0406 | 0.0431 | 0.0421 | 0.0389 | 0.0405 | 0.0441 | **0.0464*** |
| | Recall@20 | 0.0536 | 0.0478 | 0.0583 | 0.0608 | 0.0602 | 0.0650 | 0.0632 | 0.0584 | 0.0612 | 0.0661 | **0.0705*** |
| | NDCG@5 | 0.0132 | 0.0116 | 0.0150 | 0.0157 | 0.0168 | 0.0178 | 0.0160 | 0.0151 | 0.0161 | 0.0176 | **0.0191*** |
| | NDCG@10 | 0.0175 | 0.0157 | 0.0194 | 0.0206 | 0.0214 | 0.0226 | 0.0209 | 0.0198 | 0.0208 | 0.0228 | **0.0245*** |
| | NDCG@20 | 0.0225 | 0.0199 | 0.0244 | 0.0258 | 0.0263 | 0.0281 | 0.0262 | 0.0247 | 0.0260 | 0.0283 | **0.0305*** |
| Game | Recall@5 | 0.0504 | 0.0466 | 0.0578 | 0.0560 | 0.0555 | 0.0592 | 0.0572 | 0.0555 | 0.0545 | 0.0587 | **0.0616*** |
| | Recall@10 | 0.0808 | 0.0731 | 0.0926 | 0.0922 | 0.0884 | 0.0932 | 0.0917 | 0.0888 | 0.0872 | 0.0925 | **0.0972*** |
| | Recall@20 | 0.1236 | 0.1114 | 0.1392 | 0.1399 | 0.1337 | 0.1388 | 0.1381 | 0.1326 | 0.1332 | 0.1385 | **0.1444*** |
| | NDCG@5 | 0.0321 | 0.0297 | 0.0334 | 0.0343 | 0.0347 | 0.0368 | 0.0355 | 0.0341 | 0.0345 | 0.0367 | **0.0386*** |
| | NDCG@10 | 0.0419 | 0.0382 | 0.0446 | 0.0460 | 0.0452 | 0.0477 | 0.0466 | 0.0448 | 0.0450 | 0.0475 | **0.0500*** |
| | NDCG@20 | 0.0527 | 0.0478 | 0.0563 | 0.0580 | 0.0566 | 0.0592 | 0.0583 | 0.0559 | 0.0566 | 0.0591 | **0.0619*** |
| Baby | Recall@5 | 0.0204 | 0.0176 | 0.0229 | 0.0233 | 0.0231 | 0.0240 | 0.0232 | 0.0217 | 0.0220 | 0.0239 | **0.0250*** |
| | Recall@10 | 0.0338 | 0.0292 | 0.0371 | 0.0378 | 0.0368 | 0.0383 | 0.0374 | 0.0353 | 0.0357 | 0.0382 | **0.0401*** |
| | Recall@20 | 0.0548 | 0.0475 | 0.0580 | 0.0596 | 0.0576 | 0.0596 | 0.0586 | 0.0557 | 0.0571 | 0.0590 | **0.0625*** |
| | NDCG@5 | 0.0131 | 0.0112 | 0.0140 | 0.0146 | 0.0147 | 0.0153 | 0.0148 | 0.0135 | 0.0139 | 0.0146 | **0.0160*** |
| | NDCG@10 | 0.0174 | 0.0149 | 0.0186 | 0.0192 | 0.0190 | 0.0199 | 0.0193 | 0.0178 | 0.0183 | 0.0192 | **0.0208*** |
| | NDCG@20 | 0.0226 | 0.0195 | 0.0238 | 0.0247 | 0.0243 | 0.0252 | 0.0247 | 0.0230 | 0.0237 | 0.0245 | **0.0264*** |

# 3 EXPERIMENTS

In this section, we conduct extensive experiments and analysis to empirically demonstrate the effectiveness of LARES.

## 3.1 EXPERIMENT SETUP

**Dataset.** We assess our proposed method on four subsets derived from the latest Amazon 2023 review dataset: "Musical Instruments", "Video Games", "Baby Products", and "Industrial & Scientific". Following prior studies (Zhou et al., 2020; 2022; Xu et al., 2024), we employ 5-core filtering to exclude inactive users and unpopular items with fewer than five interactions to ensure a robust evaluation. All user interactions are grouped by user ID and sorted chronologically. We truncate behavior sequences to a maximum of 20 items per user. Appendix A.1 summarizes the key statistics of the preprocessed datasets.

**Baseline Models.** To conduct a comprehensive evaluation of LARES's performance, we compare it with multiple sequential recommendation baselines, including both reasoning-based and non-reasoning approaches: (1) *Non-Reasoning methods*: **GRU4Rec** (Hidasi et al., 2016a), **SASRec** (Kang & McAuley, 2018), **BERT4Rec** (Sun et al., 2019), **FMLP-Rec** (Zhou et al., 2022), **BSARec** (Shin et al., 2024), **CL4SRec** (Xie et al., 2022), **DuoRec** (Qiu et al., 2022). (2) *Reasoning-based methods*: **ERL** (Tang et al., 2025), **PRL** (Tang et al., 2025), **PRL++**. A more detailed introduction to the above baseline models is given in Appendix A.2.

**Evaluation Settings.** We evaluate the sequential recommendation task using two standard metrics: Recall@$K$ and NDCG@$K$, with $K \in \{5, 10, 20\}$. Following prior work (Zhou et al., 2020; Rajput et al., 2023; Liu et al., 2025), we adopt a *leave-one-out* strategy. Specifically, for each user interaction sequence, we use the most recent interaction as the test instance, the second-last interaction for validation, and all remaining historical interactions for training. To ensure rigorous evaluation and mitigate potential biases from negative sampling, we perform full ranking over the entire item pool. The final results report the average metric scores across all test instances.

## 3.2 OVERALL PERFORMANCE

We evaluate the performance of our proposed LARES framework by comparing it with various baseline methods across four real-world benchmark datasets. The comprehensive experimental results are presented in Table 1, from which we draw the following key observations:

Table 2: Ablation studies of LARES on three datasets. "N@K" and "R@K" denote "NDCG@K" and "Recall@K", respectively.

| Variants | Instrument | | | | Scientific | | | | Game | | | |
|---|---|---|---|---|---|---|---|---|---|---|---|---|
| | R@5 | R@10 | N@5 | N@10 | R@5 | R@10 | N@5 | N@10 | R@5 | R@10 | N@5 | N@10 |
| LAERS | **0.0411** | **0.0636** | **0.0263** | **0.0336** | **0.0293** | **0.0461** | **0.0188** | **0.0242** | **0.0616** | **0.0972** | **0.0386** | **0.0500** |
| *w/o* RPT | 0.0396 | 0.0624 | 0.0252 | 0.0326 | 0.0288 | 0.0450 | 0.0183 | 0.0235 | 0.0604 | 0.0961 | 0.0380 | 0.0491 |
| *w/o* RPT & pre-block | 0.0387 | 0.0606 | 0.0241 | 0.0311 | 0.0277 | 0.0436 | 0.0165 | 0.0216 | 0.0595 | 0.0933 | 0.0358 | 0.0467 |
| *w/o* RPT & sampling | 0.0387 | 0.0603 | 0.0250 | 0.0319 | 0.0275 | 0.0438 | 0.0175 | 0.0227 | 0.0599 | 0.0947 | 0.0370 | 0.0482 |
| *w/o* RPT & SLA | 0.0379 | 0.0596 | 0.0242 | 0.0312 | 0.0253 | 0.0403 | 0.0157 | 0.0205 | 0.0600 | 0.0943 | 0.0374 | 0.0488 |
| *w/o* RPT & SLA & TLA | 0.0356 | 0.0571 | 0.0221 | 0.0290 | 0.0245 | 0.0395 | 0.0148 | 0.0196 | 0.0559 | 0.0903 | 0.0337 | 0.0448 |

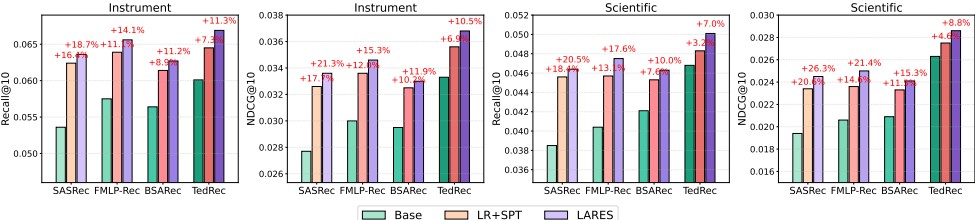

Figure 2: Performance comparison of LARES across different backbone architectures on Instrument and Scientific. 'Base' indicates the original backbone model. 'LR+SPT' refers to the backbone enhanced with our proposed latent reasoning module and pretraining. 'LARES' denotes the full implementation incorporating all proposed components.

- For non-reasoning models, filter-enhanced models (FMLP-Rec, BSARec) surpass SASRec, validating the effectiveness of frequency-domain denoising. Contrastive methods (CL4SRec, DuoRec) exceed ID-based baselines, demonstrating improved representation learning; DuoRec leads via robust model-level dropout, outperforming CL4SRec's sequence-level augmentations.

- For reasoning models, ERL achieves superior performance than SASRec across most datasets, indicating that performing multi-step reasoning can better capture the user preference. PRL consistently outperforms ERL demonstrating that noise-disturbed reasoning outcomes as contrastive signals can enhance the model's ability to extract critical sequence information. PRL++ shows significant improvements over PRL, proving that incorporating hard positive samples effectively strengthens the effectiveness of contrastive learning.

- Our proposed framework LARES outperforms all baselines, including both non-reasoning and reasoning models, by a large margin across all evaluation metrics on four datasets. This consistent superiority underscores its advanced reasoning capabilities for sequential recommendation tasks. LARES enables multi-step latent reasoning through core-block iteration at a arbitrary depth, allowing flexible computation scaling by increasing the computational density of parameters. To fully exploit its latent reasoning potential, we design two training stages, self-supervised pre-training to instill latent reasoning patterns tailored for sequential recommendation tasks and reinforcement post-training to further stimulate its reasoning capabilities by encouraging thinking exploration. These results also validate the effectiveness of depth-recurrent latent reasoning for sequential recommendation tasks.

## 3.3 Ablation Studies

We conduct an ablation study of five variants to evaluate the contribution of each key component in LARES. Specifically, we gradually remove reinforcement post-training (RPT), step-level alignment (SLA) and trajectory-level alignment (TLA). Additionally, we examine the design of architecture and step sampling by removing pre-block and fixing the reasoning step during training, respectively. From the results in Table 2, we have the following observations: (1) Removing any component tends to degrade performance confirming their necessity. (2) The combination of TLA and SLA consistently outperforms TLA alone, indicating their complementary functions: TLA ensures trajectory consistency, SLA enhances step coherence. (3) Performance drops when the pre-block module is removed, which could be due to a potential misalignment between the reasoning and item representation spaces. (4) Fixing the number of reasoning steps yields suboptimal results, possibly because it constrains the model's flexibility to explore chain-of-thought paths of varying lengths.

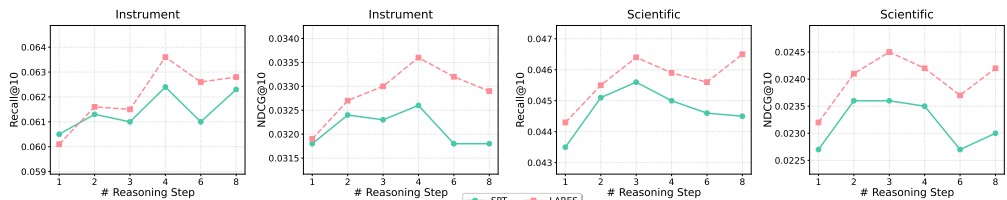

Figure 3: Performance of different reasoning steps on Instrument and Scientific. 'SPT' denotes the latent reasoning model with only pre-training. 'LARES' denotes the model with both pre-training and post-training.

### 3.4 FURTHER ANALYSIS

#### 3.4.1 COMPATIBILITY WITH DIFFERENT BACKBONES

To assess LARES's architectural compatibility, we evaluate it on three additional advanced backbones: FMLP-Rec, BSARec, and text-enhanced TedRec (Xu et al., 2024). Figure 2 compares three variants: Base, LR+SPT (latent reasoning via pretraining), and full LARES. LARES consistently enhances all models: FMLP-Rec gains 15.3% and 21.4% in NDCG@10 on Instrument and Scientific datasets; BSARec improves >10% across all metrics; even the superior TedRec benefits from LARES, achieving average NDCG@10 improvements of 10.9% and 7.9% on the Instrument and Scientific datasets. Performance consistently follows Base < LR+SPT < LARES, confirming both the compatibility of latent reasoning and the efficacy of our two-stage training. These results validate the great compatibility of our latent reasoning paradigm.

#### 3.4.2 INFLUENCE OF REASONING STEPS ON RECOMMENDATION PERFORMANCE

To analyze the influence of reasoning step, we vary the number of reasoning steps $\bar{k}$ in $\{1, 2, 3, 4, 6, 8\}$ and evaluate the model on the Instrument and Scientific datasets. The results in Figure 3 reveal a consistent trend: performance initially improves with more reasoning steps but declines after reaching an optimal point, which is in line with the findings in ReaRec. This pattern suggests that moderate reasoning step enhances model performance by strengthening representation power through additional computation. However, excessively large $\bar{k}$ leads to performance degradation, likely because simple user interaction sequences do not require intensive reasoning-a phenomenon analogous to "overthinking" in NLP (Su et al., 2025). The optimal reasoning steps are 4 for Instrument and 3 for Scientific, highlighting the importance of selecting an appropriate reasoning depth for optimal performance.

#### 3.4.3 INFLUENCE OF REINFORCEMENT POLICY OBJECTIVES

To evaluate the impact of different policy objectives in GRPO, we consider three variants of the policy $\pi(y_i|x)$: (1) Target-Only Policy: the formulation introduced in Section 2.4, which consistently models $\pi(y_i|x)$ as the joint probability of the ground-truth target item across all rollout steps; (2) Hybrid Policy: models $\pi(y_i|x)$ as the joint probability of the target item when the advantage $A_i > 0$, and as that of the top-ranked negative item when $A_i < 0$; (3) Top-Ranked Policy: treats $\pi(y_i|x)$ as the joint probability of the model's current top-ranked item at every step. The training dynamics of reward and validation performance for these variants are shown in Figure 4.

For Top-Ranked Policy, the training reward steadily decreases, suggesting a misalignment between its objective and recommendation metric. Hybrid Policy exhibits the fastest reward growth, likely because it explicitly suppresses the top negative item when $A_i < 0$. However, its validation performance fluctuates with little improvement which indicates potential overfitting from an aggressive update strategy. In contrast, Target-Only Policy yields slower reward growth but achieves more stable and consistent gains in validation metrics, confirming the effectiveness of our policy design. This stability likely stems from smoother and more consistent policy updates.

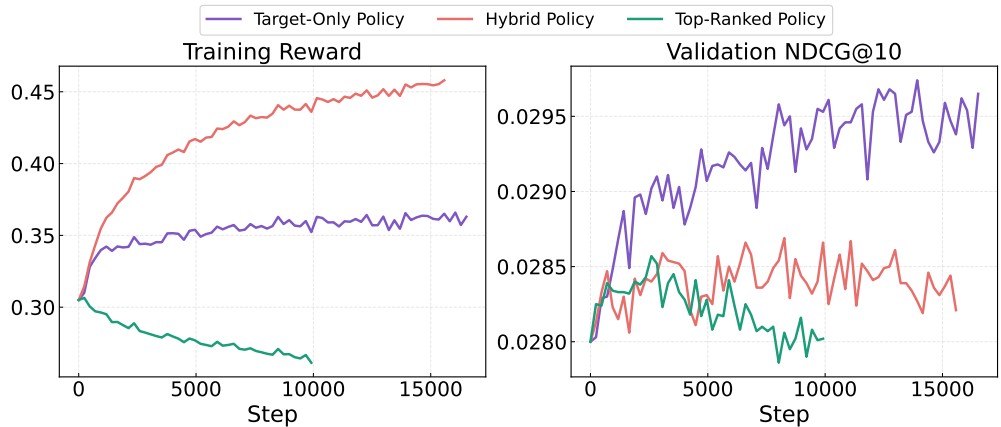

Figure 4: Performance of LARES during RPT stage on Scientific under different reinforcement learning formulations. 'Target-Only Policy' denotes modeling $\pi(y_i|x)$ as the joint probability of the target item. 'Hybrid Policy' uses the target item for $A > 0$ and the top-ranked negative item for $A < 0$. 'Top-Ranked Policy' treats $\pi(y_i|x)$ as the joint probability of the top-ranked item.

## 4 CONCLUSIONS AND LIMITATIONS

We propose **LARES**, a scalable latent reasoning framework for sequential recommendation. Unlike prior methods (*e.g.,* ReaRec), LARES utilizes all input tokens during reasoning, enhancing computational efficiency over single-token generation. Its depth-recurrent architecture comprises a pre-block and an iterative core-block, enabling test-time computation scaling without extra parameters. To maximize reasoning potential, we introduce a two-stage training pipeline: (1) Self-supervised Pre-training (SPT) with trajectory-level alignment (aligning reasoning paths sharing the same target) and step-level alignment (ensuring intra-step coherence to prevent divergence); and (2) Reinforcement Post-training (RPT), where RL encourages diverse reasoning paths using downstream metrics as rewards. Extensive experiments confirm LARES's state-of-the-art performance and seamless compatibility with modern SR architectures. However, LARES has the following limitations: (1) Computational overhead: The latent reasoning process introduces additional inference cost, despite achieving superior performance across datasets. This burden can be alleviated through model compression techniques such as knowledge distillation and quantization. (2) Hyperparameter dependence: LARES relies on hyperparameters $\alpha$ and $\gamma$ to balance different optimization objectives for optimal performance. In this work, we use LLM to polish the grammar during paper writing.

## 5 REPRODUCIBILITY STATEMENT

All results presented in this work are fully reproducible. Implementation details are provided in Appendix A.3, and hyperparameter sensitivity analyses are included in Appendix A.4. The source code is publicly available at: `https://anonymous.4open.science/r/LARES-E458/`.

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

## A SUPPLEMENT FOR EXPERIMENTS

### A.1 DATASET STATISTICS

Table 3: Statistics of the preprocessed datasets. Avg.L represents the average length of user interaction sequences.

| Dataset | #Users | #Items | #Actions | Avg.L | Sparsity |
|---|---|---|---|---|---|
| Instrument | 57,439 | 24,587 | 511,836 | 8.91 | 99.964% |
| Scientific | 50,985 | 25,848 | 412,947 | 8.10 | 99.969% |
| Game | 94,762 | 25,612 | 814,586 | 8.60 | 99.966% |
| Baby | 150,777 | 36,013 | 1,241,083 | 8.23 | 99.977% |

## A.2 Baselines

(1) *Non-Reasoning methods*:

- **GRU4Rec** (Hidasi et al., 2016a) employs GRUs to capture user behavior patterns.

- **SASRec** (Kang & McAuley, 2018) is a transformer-based model utilizing unidirectional multi-head self-attention to encode user interaction sequences.

- **BERT4Rec** (Sun et al., 2019) is a bidirectional self-attentive model that employs masked prediction for sequence modeling.

- **FMLP-Rec** (Zhou et al., 2022) replaces traditional self-attention with filter-enhanced MLPs to improve behavior modeling.

- **BSARec** (Shin et al., 2024) leverages Fourier transforms to capture both high- and low-frequency information in user behavior sequences.

- **CL4SRec** (Xie et al., 2022) first introduces contrastive learning for sequential recommendation through three augmentation strategies: item masking, reordering, and cropping.

- **DuoRec** (Qiu et al., 2022) combines unsupervised model-level dropout augmentation with hard positive sample selection.

(2) *Reasoning-based SR methods*:

- **ERL** (Tang et al., 2025) enhances sequential recommendations by aggregating multi-step implicit reasoning states.

- **PRL** (Tang et al., 2025) improves reasoning capabilities for sequential recommendation through contrastive learning with noise-disturbed positive views and temperature annealing.

- **PRL++** extends PRL by incorporating DuoRec's sampling strategy.

## A.3 Implementation Details.

We implement LARES and all baseline models using PyTorch. To ensure a fair comparison, the batch size, embedding size and hidden size are set to 1024, 64 and 256, respectively. All models are optimized using the AdamW optimizer with a learning rate of 0.001. For self-attentive models, the number of attention heads is fixed at 2. For LARES, the number of layers for both the pre-block and core-block is selected from $\{1, 2\}$, while the mean reasoning step $\bar{k}$ is chosen from $\{3, 4, 6\}$. The hyperparameter for latent reasoning state initialization $\sigma_1$ is set to 1. In SPT, we use a learning rate of 0.001 and a dropout rate of 0.5. The coefficients for trajectory-level alignment $\alpha$ and step-level alignment $\gamma$ are tuned within $\{0.1, 0.2, 0.3\}$ and $\{0.1, 0.3, 0.5, 0.7\}$, respectively. In RPT, the learning rate is selected from $\{0.0005, 0.0003, 0.0001\}$, and $\beta$ is tuned from $\{0.5, 1.0\}$. The rollout number $G$ is fixed at 4. The reward function is selected from $\{\mathrm{Recall@5}, \mathrm{Recall@10}\}$. The non-reasoning SR baselines are implemented using RecBole (Zhao et al., 2021; 2022), an open-source recommendation library, while reasoning-based SR models are reproduced from their official source codes. To mitigate overfitting, we employ early stopping, terminating training if NDCG@10 on the validation set shows no improvement for 10 consecutive epochs.

## A.4 Influence of Alignment Coefficients

We examine the effects of trajectory-level alignment and step-level alignment coefficients by evaluating model performance on Instrument and Scientific datasets across varying values $\alpha \in \{0.1, 0.2, 0.3, 0.4\}$ and $\gamma \in \{0.1, 0.3, 0.5, 0.7\}$. From the results in Figure 5, we can observe that model performance on both datasets fluctuates with increasing $\alpha$ values, peaking at 0.1 and reaching the lowest point at 0.4. This suggests that excessive trajectory-level alignment may interfere with sequential pattern learning. For $\gamma$, the performance on Instrument initially improves and then declines at higher $\gamma$ values. The performance on Scientific exhibits generally consistent improvement with increasing $\gamma$ values. These findings highlight the critical role of intra-step reasoning coherence in maintaining reasoning quality, while revealing differential sensitivity to alignment coefficients across datasets.

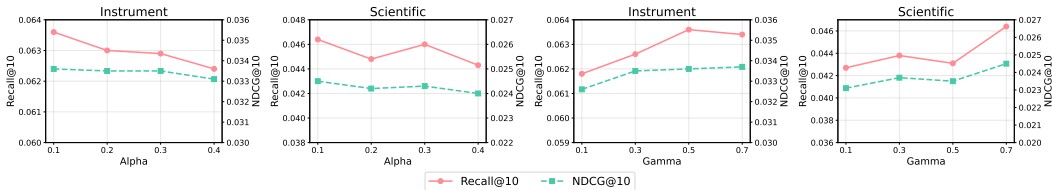

Figure 5: Performance of different alignment coefficients $\alpha$ and $\gamma$ on Instrument and Scientific.

Table 4: Inference efficiency and performance comparison across SASRec, ReaRec(PRL), and LARES under matched FLOPs on the Instrument dataset. All experiments are conducted on a single RTX 3090 GPU.

| Method | Step | Params | FLOPS | Memory | Time | Recall@5 | Recall@10 | NDCG@5 | NDCG@10 |
|---|---|---|---|---|---|---|---|---|---|
| SASRec | 1 | 0.2M | 4M | 0.57GB | 1.10s | 0.0340 | 0.0550 | 0.0209 | 0.0276 |
| | 1 | 0.3M | 6M | 0.59GB | 1.23s | 0.0339 | 0.0546 | 0.0212 | 0.0278 |
| | 1 | 0.4M | 8M | 0.61GB | 1.40s | 0.0344 | 0.0540 | 0.0215 | 0.0278 |
| | 1 | 0.5M | 10M | 0.63GB | 1.55s | 0.0338 | 0.0546 | 0.0208 | 0.0275 |
| ReaRec(PRL) | 3 | 0.2M | 4.4M | 4.76GB | 1.21s | 0.0345 | 0.0551 | 0.0222 | 0.0288 |
| | 3 | 0.3M | 6.6M | 6.92GB | 1.44s | 0.0346 | 0.0549 | 0.0215 | 0.0280 |
| | 3 | 0.4M | 8.8M | 9.21GB | 1.71s | 0.0342 | 0.0544 | 0.0213 | 0.0277 |
| | 3 | 0.5M | 11M | 11.35GB | 1.94s | 0.0344 | 0.0547 | 0.0216 | 0.0278 |
| LARES | 1 | 0.2M | 4M | 0.55GB | 1.19s | 0.0384 | 0.0601 | 0.0249 | 0.0319 |
| | 2 | 0.2M | 6M | 0.57GB | 1.39s | 0.0395 | 0.0616 | 0.0256 | 0.0327 |
| | 3 | 0.2M | 8M | 0.59GB | 1.58s | 0.0399 | 0.0615 | 0.0260 | 0.0330 |
| | 4 | 0.2M | 10M | 0.61GB | 1.74s | **0.0411** | **0.0636** | **0.0263** | **0.0336** |

## A.5 INFLUENCE OF REASONING STEPS ON INFERENCE EFFICIENCY AND PERFORMANCE

To systematically assess the impact of reasoning steps on inference efficiency and performance, we conduct a comparative analysis across LARES, ReaRec(PRL) and SASRec under matched computational budgets on Instrument. All models use a batch size of 1024; LARES employs a 2-layer pre-block and a 2-layer core-block. As shown in Table 4 (item embeddings excluded from parameter counts), LARES exhibits superior scalability: performance steadily improves with more reasoning steps. In contrast, SASRec and ReaRec saturate as model depth increases—likely due to insufficient data for effective optimization. Notably, LARES can achieve higher computing power without extra parameters through increased reasoning steps. For instance, with 4 reasoning steps, LARES attains an effective depth of 2+2×4 layers, comparable to a 10-layer SASRec architecture. The additional inference latency introduced by LARES remains moderate—approximately 10% higher than SASRec at comparable FLOPs. This computational overhead is acceptable in consideration of LARES's substantial performance gains, which improve SASRec's baseline by an average of 20% on Instrument as shown in Figure 2. In terms of GPU memory consumption, we observe that LARES incurs slightly less GPU memory overhead than SASRec at the same level of FLOPs, owing to its computing scaling ability with fixed parameters. In contrast, ReaRec imposes substantially higher GPU memory demands, which stems from its KV cache mechanism to boost inference efficiency. Our findings demonstrate that LARES offers an advantageous trade-off between computational cost and model performance, suggesting strong potential for practical deployment.

## A.6 INFLUENCE OF REINFORCEMENT REWARDS

We evaluate the impact of reinforcement rewards during RPT stage through ablation studies on Recall@$K$ and NDCG@$K$ with $K \in \{5, 10, 20\}$. As shown in Figure 6, most reward variants improve the pretrained model, as they are inherently aligned with recommendation objectives. However, we observe a decline in performance on fine-grained evaluation metrics such as R@$\{5,10\}$ and N@$\{5,10\}$ when the reward parameter $K$ is increased from 10 to 20. This degradation likely stems from the fact that rewards based on $K = 20$ fail to capture subtle ranking differences among the top-20 items. Therefore, it is crucial to select reward signals with appropriate granularity based on the scenario requirements.

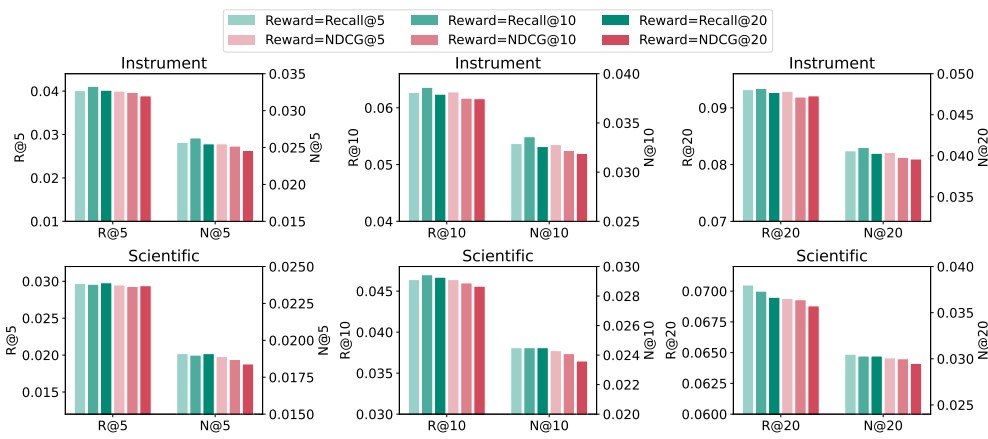

Figure 6: Performance of different rewards on Instrument and Scientific.

## B    TIME COMPLEXITY

As described in Section 2.2, the LARES framework comprises two key components: a pre-block and a core-block. To illustrate the computational complexity, we adopt the transformer architecture as a representative example due to LARES's compatibility with diverse model designs. The primary computational overhead in each transformer layer stems from the multi-head self-attention, with complexities of $\mathcal{O}(N^2d + Nd^2)$, where $N$ represents the sequence length and $d$ denotes the model dimension. Let $L_1$ and $L_2$ denote the number of transformer layers in the pre-block and core-block, respectively. Consequently, their computational complexities can be expressed as $\mathcal{O}(L_1(N^2d+Nd^2))$ and $\mathcal{O}(L_2(N^2d + Nd^2))$. For a reasoning process involving $K$ iterative steps, the overall time complexity of LARES scales as $\mathcal{O}((L_1 + L_2K)(N^2d + Nd^2))$.

## C    RELATED WORK

**Sequential Recommendation** Sequential recommendation has become a prominent research area in recommender systems, with the objective of modeling latent patterns in user behavior sequences to predict the next item of interest. Early approaches (Rendle et al., 2010; He & McAuley, 2016) modeled user behaviors as Markov chains, focusing exclusively on item transition patterns. The advent of deep learning revolutionized this field, leading to the adoption of various neural architectures. These include convolutional neural networks (CNNs) (Tang & Wang, 2018; Yan et al., 2019; Yuan et al., 2019), recurrent neural networks (RNNs) (Hidasi et al., 2016a; Tan et al., 2016; Hidasi et al., 2016b; Quadrana et al., 2017), and graph neural networks (GNNs) (Chang et al., 2021; Wu et al., 2019). Recently, Transformer-based models (Kang & McAuley, 2018; Sun et al., 2019; Hao et al., 2023) have demonstrated superior performance in sequential behavior modeling. Several recent studies (Chen et al., 2022; Wang et al., 2023; Zhou et al., 2023; Liu et al., 2024) have proposed to enhance Transformer architectures. FMLP-Rec (Zhou et al., 2022) replaces self-attention with filter-enhanced MLPs to reduce noise in user preference modeling. However, these ID-based methods often suffer from cold-start problems. To address this, alternative approaches incorporate item textual metadata to enrich representations (Hou et al., 2022; Xu et al., 2024; Liu et al., 2025). UniSRec employs multi-domain textual data with MoE-enhanced adapters to learn universal sequential representations. TedRec achieves sequence-level fusion of textual and ID representations through contextual convolution. Despite these advancements, the test-time scaling in sequential recommendation is underexplored. In this paper, we propose a new latent reasoning paradigm for sequential recommendation that leverages all input tokens to perform multi-step reasoning in latent space with arbitrary depth.

**Reasoning Models** Recent advances in test-time scaling (DeepSeek-AI et al., 2025; Chen et al., 2025b;a) have shifted the research focus in GenAI from large language models (LLMs) to large

reasoning models (LRMs). LRMs excel in complex reasoning tasks through deep thinking capabilities, as demonstrated by the powerful reasoning systems like OpenAI-o1 and DeepSeek-R1. These models employ explicit long Chain-of-Thought mechanisms to generate extensive reasoning tokens before producing final answers (Yang et al., 2024; Besta et al., 2024; Yang et al., 2025). However, their reliance on explicit reasoning poses some critical challenges, including excessive memory demands from long context windows and limited expressive power due to discrete language space constraints. To address these limitations, recent studies (Hao et al., 2024; Chen et al., 2024; Geiping et al., 2025; Xu et al., 2025b) have introduced latent reasoning models that perform implicit reasoning in continuous latent spaces, achieving greater efficiency. In recommender systems, some efforts have adapted reasoning mechanisms for sequential recommendation models (Tang et al., 2025; Zhang et al., 2025). For instance, ReaRec (Tang et al., 2025) autoregressively generates latent reasoning tokens to refine user representations. STREAM-Rec (Zhang et al., 2025) integrates slow-thinking paradigms with TIGER-style generative recommenders by producing annotated reasoning tokens before final recommendation semantic tokens. In contrast, we propose **LARES**, a novel recurrent-depth latent reasoning framework for sequential recommendation. Unlike prior work, LARES iteratively refines all input tokens at each reasoning step. Additionally, it is seamlessly compatible with existing sequential recommendation models, further enhancing their performance.

