# OpenReview forum: "Latent Reasoning with Recurrent Depth for Sequential Recommendation"
_ICLR.cc/2026/Conference — Submitted to ICLR 2026_

### Official Review · Reviewer_K6R5 · 2025-10-28

**Soundness:** 3
**Presentation:** 3
**Contribution:** 2
**Rating:** 4
**Confidence:** 4

**Summary:**

LARES introduces a depth-recurrent latent-reasoning module for sequential recommendation: a pre-block projects sequences to a latent space, and a core block refines all token states over K steps. Training is two stage, with self-supervised pre-training (TLA, SLA) followed by RL post-training (GRPO-style) optimizing ranking rewards. On four Amazon-2023 subsets it improves over strong baselines, supported by ablations and a latency/FLOPs study.

**Strengths:**

Solid empirical gains across multiple approaches (SASRec, FMLP-Rec, BSARec, TedRec).

Practical deployment angle: controllable test-time computation; latency/FLOPs study.

Thoughtful ablations and step-count study; the “overthinking” dip is informative.

**Weaknesses:**

The theoretical justification is weak. What is "reasoning" in recommendation? How does it differ from just deeper models?

It appears that the TLA constraint contradicts the main claim. It can't handle variable-length paths in training, but claims flexible test-time scaling

The RL design is weak. Simple rewards, filters hard examples.

Please see questions for more specific points.

**Questions:**

Please explicitly define the "reasoning step" formally for recommendations. The "reasoning" appears to be iterative refinement of representations in latent space, not reasoning about behavioral motivations. I'm unclear on what reasoning actually means here.

RL formalization: please define state, action, policy, reward, advantage, and the exact KL with the reference policy; show how gradients flow (p. 6, Eq. 11–12 and the reformulation paragraph).

Pruning impact: what share of training instances is dropped by the top-100 filter per dataset? Please report results with no pruning and with hard cases.

Why require matched steps across positives?

Table 4: what exactly do “2x+2x” and “4(1+δ)x” denote, and how are FLOPs “matched” across different kernels/memory patterns?

TLA treats sequences sharing the same target item as positives. Provide robustness by item-popularity decile and an alternative that controls for item identity. Is there popularity leakage?

Please show intermediate-step probes (entropy, calibration, retrieval accuracy by step), and variance of latent states over steps to demonstrate SLA prevents collapse.

---

> ### Author Response · Authors · 2025-11-24
> **Official Response (Part 1)**
>
> Thank you for your detailed review. We have addressed the points you mentioned as follows.
>
> > **Q1: Definition of reasoning step in recommendation**
>
> A1: We follow the definition of latent reasoning in work [1]. **In our response to Q3 from Reviewer eZEV**, we further clarify the notion of reasoning and its conceptual relationship to explicit chain-of-thought and latent reasoning. We believe that **reasoning should not be limited to human language space**; rather, language simply provides a more interpretable and supervised way for humans to observe the reasoning process.
>
> [1] Scaling up Test-Time Compute with Latent Reasoning: A Recurrent Depth Approach. arXiv 2025.
>
> > **Q2: Clarification of RL formalization**
>
> A2: Here we give a clearer definiton of the RL formulation.
>
> The state consists of the input historical user sequence $S_u$ and previously generated latent state $t_{i-1}$.
>
> The action is the output latent state $t_i$ produced by the model.
>
> The policy is a conditional distribution over actions given states $\pi_\theta(t_i \mid S_u,t_{i-1})$.
>
> Since $\pi(t_i \mid S_u,t_{i-1})$ is infeasible to compute directly, thus, we define the **joint probability of a rollout** as the likelihood of recommending the correct next item at every step $\pi_\theta(y_i \mid x)=\prod_{j=1}^{k}
> \pi_\theta(v_{n+1} \mid S_u, t_{i,j})$.
>
> Reward $r$ is the ranking metrics such as Recall@10. The advantage is defined as the improvement of a rollout relative to others in the group: $A_i = \frac{r_i - \operatorname{mean}(r_1,\ldots,r_G)}{\operatorname{std}(r_1,\ldots,r_G)}$.
>
> The KL divergence is computed as:
>
> $$
> D_\text{KL}\left(\pi_\theta||\pi_{{\text{ref}}}\right)=\prod_{j=1}^k\frac{\pi_{{\text{ref}}}(v_{n+1}|S_u,t_{i,j})}{\pi_{\theta}(v_{n+1}|S_u,t_{i,j})}-\sum_{j=1}^k\log\frac{\pi_{{\text{ref}}}(v_{n+1}|S_u,t_{i,j})}{\pi_{\theta}(v_{n+1}|S_u,t_{i,j})}-1
> $$
>
> To summarize, the objective of the RL formulation is to increase the probability of the target item appearing in the top-k recommendation list.
>
> In response to Q1 from Reviewer Z3f7, we further discuss the design of our RL formulation and present the **training dynamics in Section 3.4.3** of the revised version. As training progresses, we observe that both the reward and validation NDCG@10 increase, demonstrating the effectiveness of our RL design.
>
> > **Q3: Impact of data selection**
>
> A3: In response to **Q4 from Reviewer Z3f7**, we conducted experiments to evaluate the impact of data selection. The results show that the data selection strategy significantly **reduces training time while simultaneously improving final performance**, demonstrating its effectiveness.
>
> > **Q4: Why require matched steps across positives**
>
> A4: In our experiments, we observed that **matching reasoning steps** in TLA yields better performance than unmatched steps (see Table 8). A likely reason is that **misalignment between short-chain and long-chain reasoning** can degrade long-chain performance. Specifically, forcing short-chain reasoning to capture the richer and more complex patterns of longer chains is inherently challenging. This often causes the model to **degenerate toward short-chain reasoning**, ultimately reducing its overall effectiveness.
>
> **Table 8. Performance for unmatched steps alignment.**
>
> |               | Instrument |            |            |            | Scientific |            |            |            |
> | ------------- | ---------- | ---------- | ---------- | ---------- | ---------- | ---------- | ---------- | ---------- |
> |               | Recall@5   | Recall@10  | NDCG@5     | NDCG@10    | Recall@5   | Recall@10  | NDCG@5     | NDCG@10    |
> | LARES (SPT)   | **0.0396** | **0.0624** | **0.0252** | **0.0326** | **0.0288** | **0.0450** | **0.0183** | **0.0235** |
> | w/o same step | 0.0382     | 0.0609     | 0.0234     | 0.0307     | 0.0284     | 0.0444     | 0.0171     | 0.0222     |

---

> ### Author Response · Authors · 2025-11-24
> **Official Response (Part 2)**
>
> > **Q5: What do "2x+2x" and "4(1+δ)x" denote, and how are FLOPs “matched” across different kernels/memory patterns?**
>
> A5: In this context, "x" represents the total number of parameters in one transformer layer. For LARES, the model consists of two transformer layers in the pre-block and two transformer layers in the core-block. Therefore, the total number of parameters in LARES is "2x + 2x".
>
> For the FLOPs, we use "x" to approximate the number of FLOPs required for a single transformer layer to process one sequence. Hence, for a transformer with four layers, the total FLOPs are "4x".
>
> ReaRec follows an autoregressive latent reasoning pattern, which generates $k$ additional hidden states. Taking into account the kv cache mechanism, the extra computational cost is approximately linear with respect to the initial cost. Here, "δ" represents this ratio. Thus, the approximate FLOPs for ReaRec with four layers is "4(1+δ)x".
>
> To clarify, we have updated **Table 4 in Appendix A.5** to show the actual parameters and FLOPs. All experiments were conducted in the same software and hardware environment to ensure a fair comparison.
>
> > **Q6: Provide robustness by item-popularity decile and an alternative that controls for item identity.**
>
> A6: To evaluate the influence of item popularity, we restricted the positives in TLA to the same sequence and observed a performance decline, as shown in Table 8 below. This indicates that limiting the diversity of cross-sequence alignment would reduce the effectiveness of TLA. We think one reason for TLA's effectiveness is that it alleviates popularity bias by **aligning positive items with different popularity levels, making the overall representation space more uniform**, as supported by prior work [1].
>
> **Table 9. Ablation studies for different positive samples in TLA.**
>
> |                      | Instrument    |               |               |               | Scientific    |               |               |               |
> | -------------------- | ------------- | ------------- | ------------- | ------------- | ------------- | ------------- | ------------- | ------------- |
> |                      | Recall@5      | Recall@10     | NDCG@5        | NDCG@10       | Recall@5      | Recall@10     | NDCG@5        | NDCG@10       |
> | LARES (TLA)          | **0.0379**    | **0.0596**    | **0.0242**    | **0.0312**    | **0.0271**    | **0.0414**    | **0.0173**    | **0.0219**    |
> | w/ sequence identity | 0.0367 | 0.0584 | 0.0231 | 0.0302 | 0.0258 | 0.0404 | 0.0161 | 0.0207 |
> | LR only              | 0.0356        | 0.0571        | 0.0221        | 0.0290        | 0.0245        | 0.0395        | 0.0148        | 0.0196        |
>
> [1] Contrastive Learning for Representation Degeneration Problem in Sequential Recommendation. WSDM 2022.
>
> > **Q7: Please show intermediate-step probes (entropy, calibration, retrieval accuracy by step), and variance of latent states over steps to demonstrate SLA prevents collapse.**
>
> A7: In **Table 2 of the paper**, we present ablation studies that evaluate the effect of SLA. Specifically, the results of the variants "**w/o RPT & SLA**" and "**w/o RPT**" highlight the significant performance improvements brought by SLA.
>
> To further assess its impact, we provide the **per-step retrieval accuracy and similarity** for LARES and LARES without SLA on the Instrument in Table 10 and 11 below. From these results, we observe that LARES' performance **continues to improve as reasoning steps deepen**, whereas **LARES without SLA quickly saturates after step 2 with the cosine similarity between adjacent reasoning steps reaching 1 after step 2.** This demonstrates that SLA enhances the reasoning process by maintaining coherence across steps, preventing early performance stagnation and model collapse.
>
> **Table 10. Recall@10 at different reasoning steps for SLA variants on Instrument**
>
> | Steps   | 1          | 2          | 3          | 4          |
> | ------- | ---------- | ---------- | ---------- | ---------- |
> | LARES   | **0.0595** | 0.0603     | **0.0627** | **0.0636** |
> | w/o SLA | 0.0578     | **0.0610** | 0.0607     | 0.0607     |
>
> **Table 11. Cosine similarity between adjacent reasoning steps for SLA variants on Instrument**
>
> | Steps   | 1-2      | 2-3      | 3-4      |
> | ------- | -------- | -------- | -------- |
> | LARES   | 0.89     | 0.93     | 0.96     |
> | w/o SLA | **0.92** | **1.00** | **1.00** |
>
> > **Q8: It appears that the TLA constraint contradicts the main claim. It can't handle variable-length paths in training, but claims flexible test-time scaling**
>
> A8: This may be a misunderstanding. In TLA, we only require **positive samples within the same batch** to have matched reasoning steps; **steps can vary across batches**. In practice, we **randomly sample reasoning steps for each batch**, thus LARES can achieve flexible test-time scaling.

---

> ### Author Response · Authors · 2025-11-28
> **Looking forward to your reply!**
>
> Dear Reviewer K6R5,
>
> Thank you for your time and thoughtful feedback on our paper. We sincerely appreciate the effort you’ve taken to review our work. We have carefully addressed all of your concerns in our response and hope it meet your expectations.
>
> If you have any further suggestions or questions, please don’t hesitate to let us know.

---

### Official Review · Reviewer_1a8J · 2025-11-01

**Soundness:** 3
**Presentation:** 3
**Contribution:** 3
**Rating:** 6
**Confidence:** 3

**Summary:**

The paper introduces LARES, a latent-reasoning framework for sequential recommendation that deepens a model’s “thinking” by repeatedly applying a compact reasoning module—i.e., recurrent depth—to increase the computational density of existing parameters rather than adding new ones. This depth-recurrent design flexibly scales reasoning without extra parameters, enabling richer modeling of complex, evolving user interests. LARES is trained in two stages: (1) Self-supervised pre-training with both trajectory-level and step-level alignment to learn task-specific latent reasoning signals without annotations; and (2) Reinforcement post-training to explore diverse reasoning paths and refine decision quality. Experiments on real-world benchmarks show consistent gains and plug-and-play compatibility with strong baselines.

**Strengths:**

Overall, the paper is strong across originality, quality, clarity, and significance. It is original in framing latent reasoning with recurrent depth as a compute-scaling mechanism for sequential recommendation, innovatively increasing computational density without adding parameters and pairing this with tailored trajectory- and step-level self-supervision plus reinforcement post-training that optimizes ranking-metric rewards. The empirical quality is high: results on real-world benchmarks use full-ranking evaluation, show consistent and statistically reliable gains, include careful ablations and reasoning-depth analyses, and demonstrate plug-and-play improvements across multiple backbones. The exposition is clear, with a coherent narrative from motivation to architecture and training objectives, precise mathematical specification, informative figures, and explicit discussion of limitations such as compute overhead and hyperparameter sensitivity. In significance, the approach offers a practically attractive path to “deeper thinking” via test-time compute rather than parameter growth, increases the likelihood of deployment under memory budgets, and helps bridge LLM-style reasoning paradigms into recommendation, setting the stage for broader follow-up work on latent reasoning in discrete-ID tasks.

**Weaknesses:**

- The paper does not evaluate on more complex, large-scale benchmarks such as Tmall or Yelp, and it does not include long horizon behavior datasets such as MovieLens 1M or MovieLens 20M. Without these results, it is difficult to judge scalability to dense catalogs and the ability to model very long user histories.

- The paper does not compare with graph-based recommendation methods that model high-order item transitions and relations through message passing or more recent graph contrastive sequence models. This gap weakens the empirical claim of broad superiority. I recommend expanding the related work to systematically review sequential recommendation across attention, convolutional, recurrent, state space, and graph-based families, and to clearly state which limitations LARES addresses in each line. It would also help to add a focused discussion of large model-based recommendation, including prompt-driven and adapter-style approaches, recent work on latent reasoning and test-time compute scaling for recommendation, and to include representative LLM baselines in the experiments.

- A major weakness is the absence of a direct comparison of the reinforcement learning stage’s training efficiency against strong non-RL alternatives under matched compute and data. It is therefore unclear whether reinforcement learning brings distinctive benefits in recommendation beyond well tuned supervised objectives. The paper should report learning curves of Recall and NDCG versus wall clock time and versus number of consumed examples, provide compute normalized comparisons to pairwise and listwise losses, and include policy gradient–free preference optimization such as DPO style ranking, as well as off policy bandit training with IPS or doubly robust estimators.

- Training stability, variance across seeds, and convergence speed should be documented, together with sensitivity to reward scale and KL control. The authors should also delineate when reinforcement learning helps, for example under sparse delayed feedback, exposure bias, or longer horizon objectives, and when it does not. Finally, demonstrate combinations that may outperform pure reinforcement learning, such as supervised pretraining followed by short RL refinement, multi objective training that mixes listwise cross entropy with the RL reward, or using reinforcement learning only for a depth controller while keeping the backbone trained with supervised losses. This evidence would clarify whether reinforcement learning is uniquely advantageous in specific recommendation regimes or primarily an effective complement to standard training.

**Questions:**

- How does LARES perform on complex, large scale datasets such as Tmall or Yelp, and on long horizon histories such as MovieLens 1M and 20M？

- How does LARES compare to graph based sequence models that capture high order transitions with message passing and recent graph contrastive approaches？

- What is the relative benefit of LARES versus prompt-driven or adapter-based large model recommenders？

- What is the cost of deeper reasoning at inference in terms of FLOPs, wall clock latency at median and tail, and throughput？

- Under what regimes does reinforcement learning help the most, for example sparse delayed feedback, severe exposure bias, or longer horizon goals？

- What do intermediate reasoning states capture and how do they change the ranked list？

---

> ### Author Response · Authors · 2025-11-24
> **Official Response (Part 1)**
>
> Thank you for your thoughtful feedback. We’ve carefully reviewed your comments and would like to address the following points.
>
> > **Q1: Performance on large scale datasets and long horizon histories**
>
> **A1:** We conduct experiments on two additional datasets, yelp2018 and movieLens-1m (ml-1m), to evaluate LARES in large-scale and long-history settings. Both datasets are preprocessed using 5-core filtering, resulting in 213,170 users, 94,304 items, and 3,277,932 interactions for yelp2018, and 6,040 users, 3,416 items, and 999,611 interactions for ml-1m. We set the maximum sequence length to 20 for yelp2018 and 100 for ml-1m. And we use the leave-one-out evaluation strategy, and other settings remain consistent with those in the main paper.
>
> Results are reported in Table 5. LARES consistently outperforms all baselines across all metrics, demonstrating its effectiveness in both large-scale and long-horizon recommendation scenarios.
>
> **Table 5. Performance on ml-1m and yelp2018**
>
> | Dataset | ml-1m       |             |             |             | yelp2018    |             |             |             |
> | ------- | ----------- | ----------- | ----------- | ----------- | ----------- | ----------- | ----------- | ----------- |
> | Model   | Recall@10   | Recall@20   | NDCG@10     | NDCG@20     | Recall@10   | Recall@20   | NDCG@10     | NDCG@20     |
> | SASRec  | 0.2389      | 0.3518      | 0.1203      | 0.1499      | 0.0443      | 0.0737      | 0.0216      | 0.0290      |
> | FMLPRec | 0.2399      | 0.3548      | 0.1266      | 0.1555      | 0.0455      | 0.0762      | 0.0221      | 0.0298      |
> | DuoRec  | 0.2538      | 0.3672      | 0.1309      | 0.1608      | 0.0469      | 0.0763      | 0.0232      | 0.0306      |
> | ReaRec  | 0.2386      | 0.3608      | 0.1225      | 0.1521      | 0.0450      | 0.0752      | 0.0220      | 0.0295      |
> | LARES   | **0.2621*** | **0.3696*** | **0.1358*** | **0.1667*** | **0.0493*** | **0.0795*** | **0.0241*** | **0.0318*** |
>
>
>
> > **Q2: Comparison with graph based contrastive sequence models**
>
> A2: We compare LARES with two representative graph-enhanced contrastive sequential recommendation models GCL4SR[1] and MAERec[2] on Instrument and Scientific. Upon examining their official codes, we find that they both mask all history items during the evaluation stage, which has a significant impact on the final performance. **To ensure a fair and comprehensive comparison, we conduct experiments under both settings (mask and not mask)** and the results are shown in Table 6 and 7, respectively. In both settings, LARES consistently achieves higher performance across all metrics compared to GCL4SR and MAERec. This demonstrates that the latent reasoning mechanism in LARES is effective in capturing sequential user intent.
>
> **Table 6. Performance on Instrument and Scientific (without mask historical items)**
>
> |        | instrument  |             |             |             | scientific  |             |             |             |
> | ------ | ----------- | ----------- | ----------- | ----------- | ----------- | ----------- | ----------- | ----------- |
> |        | Recall@5    | Recall@10   | NDCG@5      | NDCG@10     | Recall@5    | Recall@10   | NDCG@5      | NDCG@10     |
> | GCL4SR | 0.0354      | 0.0559      | 0.0229      | 0.0295      | 0.0249      | 0.0374      | 0.0163      | 0.0203      |
> | MAERec | 0.0325      | 0.0522      | 0.0210      | 0.0273      | 0.0232      | 0.0370      | 0.0147      | 0.0191      |
> | LARES  | **0.0411*** | **0.0636*** | **0.0263*** | **0.0336*** | **0.0297*** | **0.0464*** | **0.0191*** | **0.0245*** |
>
> **Table 7. Performance on Instrument and Scientific (mask historical items)**
>
> |        | instrument  |             |             |             | scientific  |             |             |             |
> | ------ | ----------- | ----------- | ----------- | ----------- | ----------- | ----------- | ----------- | ----------- |
> |        | Recall@5    | Recall@10   | NDCG@5      | NDCG@10     | Recall@5    | Recall@10   | NDCG@5      | NDCG@10     |
> | GCL4SR | 0.0391      | 0.0593      | 0.0263      | 0.0328      | 0.0289      | 0.0402      | 0.0201      | 0.0237      |
> | MAERec | 0.0369      | 0.0577      | 0.0243      | 0.0310      | 0.0303      | 0.0452      | 0.0205      | 0.0253      |
> | LARES  | **0.0443*** | **0.0665*** | **0.0294*** | **0.0366*** | **0.0330*** | **0.0499*** | **0.0224*** | **0.0278*** |
>
> [1] Enhancing Sequential Recommendation with Graph Contrastive Learning. IJCAI 2022.
>
> [2] Graph Masked Autoencoder for Sequential Recommendation. SIGIR 2023.

---

> ### Author Response · Authors · 2025-11-24
> **Official Response (Part 2)**
>
> > **Q3: What is the relative benefit of LARES versus prompt-driven or adapter-based large model recommenders？**
>
> A3: Compared to prompt-driven and adapter-based LLM recommenders, LARES offers a key advantage in **performance, training and inference efficiency**. Specifically:
>
> - Efficiency: LARES is a latent reasoning-enhanced framework built on pure recommendation backbones, **enabling flexible control of computing latency through adjustable inference depth**. At matched inference depths, LARES has the same computational complexity as SASRec.
> - Scaling performance: Under equivalent FLOPS, LARES shows better scaling performance than its backbone (see **Appendix A.5**).
>
> While LLM-based recommenders benefit from rich world knowledge and strong semantic understanding, they require **substantial computational resources** and incur **heavy inference latency**, making them impractical for large-scale deployment. Furthermore, it remains unclear whether using LLMs as the core recommendation engine represents the future of recommender systems, and further investigation is needed.
>
> Importantly, LARES can effectively **leverage knowledge distilled from LLMs** (such as user and item representations) while avoiding their **substantial computational overhead**.
>
> To further support our claims, we compare LARES with LatentR3 based on Qwen2.5-1.5B-Instruct in **Table 3**. Please refer to our response to **Q2 from Reviewer Z3f7** for details.
>
> > **Q4: What is the cost of deeper reasoning at inference in terms of FLOPs, wall clock latency at median and tail, and throughput？**
>
> A4: The FLOPs associated with deeper reasoning during inference are equivalent to those of SASRec with the same inference depth. For more details, **please refer to the latency experiments in Appendix A.5**.
>
> > **Q5: Under what regimes does reinforcement learning help the most, for example sparse delayed feedback, severe exposure bias, or longer horizon goals？**
>
> A5: In this work, we demonstrate that **reinforcement post-training is effective in enhancing latent reasoning ability without requiring annotated reasoning data** in our settings. While investigating the impact of RL under different regimes (such as severe exposure bias, or longer horizon goals) is an important direction, it is a bit beyond the scope of this paper. Additionally, we have not yet conducted experiments in online industrial environments, so we are unable to draw definitive conclusions in these settings at this time.
>
> > **Q6: What do intermediate reasoning states capture and how do they change the ranked list？**
>
> A6: In LARES, the model performs reasoning in a continuous latent space to better model user behavior and generate recommendations. **The intermediate reasoning states capture user preference information from surface-level to deeper insights**. Each of these states can be used to match items and make recommendations. In our response to **Q3 from Reviewer eZEV**, we present the performance results of outputs at different steps within the same forward pass. It is evident that **as the reasoning deepens, performance continues to improve**, demonstrating that the model is actively thinking in the latent space to better capture user preferences.
>
> > **Q7: The paper should report learning curves of Recall and NDCG versus wall clock time and versus number of consumed examples**
>
> A7: We provide the learning curves of **rewards and NDCG** during RPT in **Section 3.4.3 of the revised version**. The curves show that both rewards and NDCG **consistently improve with increasing training steps**, demonstrating the effectiveness of our approach.

---

> ### Author Response · Authors · 2025-11-28
> **Looking forward to your reply!**
>
> Dear Reviewer 1a8J,
>
> Thank you for your time and thoughtful feedback on our paper. We sincerely appreciate the effort you’ve taken to review our work. We have carefully addressed all of your concerns in our response and hope it meet your expectations.
>
> If you have any further suggestions or questions, please don’t hesitate to let us know.

---

### Official Review · Reviewer_Z3f7 · 2025-11-01

**Soundness:** 2
**Presentation:** 3
**Contribution:** 3
**Rating:** 6
**Confidence:** 5

**Summary:**

This paper proposes LARES, a scalable latent reasoning framework for sequential recommendation, aiming to address the limitations of current non-reasoning paradigms and parameter scaling methods. LARES adopts a depth-recurrent architecture consisting of a pre-block and an iterable core-block, enabling multi-step latent reasoning with all input tokens without additional parameters. To unlock its reasoning potential, a two-stage training strategy is designed: Self-supervised Pre-training (SPT) with trajectory-level and step-level alignment, and Reinforcement Post-training (RPT) leveraging GRPO algorithm. Extensive experiments on four Amazon datasets demonstrate LARES outperforms existing non-reasoning and reasoning baselines, with seamless compatibility with advanced sequential recommendation backbones.

**Strengths:**

## 1. Originality.
The depth-recurrent latent reasoning paradigm is innovative, differing from prior work (e.g., ReaRec) that only generates single tokens per reasoning step. LARES refines all input tokens iteratively, effectively improving computational density and reasoning efficiency.

##  2. Quality
Experimental design is rigorous: four real-world datasets and comprehensive baselines ensure fair comparison.
Ablation studies systematically verify the necessity of core components (pre-block, SLA, TLA, RPT), and additional analyses (reasoning steps, alignment coefficients, inference efficiency) deepen the understanding of the framework.

## 3. Significance
LARES's compatibility with existing backbones enables easy integration into practical systems, promoting industrial application.

**Weaknesses:**

## 1.  Insufficient Discussion on Computational Overhead Trade-offs.

Although the paper identifies computational overhead as a limitation, it fails to conduct quantitative analysis in practical application scenarios. This makes it impossible to clearly demonstrate key performance metrics of the framework such as inference latency and memory usage.  Additionally, there is no horizontal comparison of computational efficiency with mainstream baselines, leaving readers unable to evaluate its feasibility and cost trade-offs in real-world deployments.

## 2. Inadequate Comparison with Baselines.
The concurrent work Reinforced Latent Reasoning for LLM-based Recommendation (LatentR3) shares a highly similar technical route with this paper: both leverage reinforcement learning to implement latent reasoning mechanisms for sequential recommendation. However, the paper does not include LatentR3 in comparative experiments, which prevents a clear illustration of the framework’s advantages in core designs and its potential for performance improvement, thereby weakening the persuasiveness of the research conclusions.

## 3. Incomplete Ablation Experiments for Reinforcement Learning Components.
The data selection strategy adopted in the paper lacks targeted ablation validation. No control group experiment with "retained hard samples (samples where the target item ranks below 100)" is designed, making it impossible to verify the specific impact of the current filtering strategy on the model’s generalization ability and reasoning accuracy.

**Questions:**

## 1. Technical Questions
- In the GRPO adaptation, why is the joint probability reformulated as the product of target item recommendation probabilities at each step? Is there theoretical support for this approximation?

- Your work and the concurrent study Reinforced Latent Reasoning for LLM-based Recommendation (LatentR3) both integrate reinforcement learning with latent reasoning for sequential recommendation. Could you provide a detailed comparative analysis between LARES and LatentR3?

## 2. Experimental Suggestions
- Supplement computational overhead experiments: Test inference latency and memory usage under different sequence lengths and batch sizes, and compare with baselines under the same hardware conditions.

- Complete RPT component ablation: Compare the performance of different reward functions and data selection strategies to demonstrate the necessity of the proposed design.

**Details Of Ethics Concerns:**

No concern

---

> ### Author Response · Authors · 2025-11-24
> **Official Response (Part 1)**
>
> We appreciate your detailed insights. After considering your concerns, we’d like to offer our responses below.
>
> > **Q1: Why is the joint probability reformulated as the product of target item recommendation probabilities at each step?**
>
> A1: In the original GRPO formulation, $\pi(O|q)$ denotes the joint probability of a rollout trajectory, encouraging higher likelihood for correct trajectories and lower likelihood for incorrect ones. In recommendation, this naturally corresponds to **increasing the probability of the next target item**.
>
> Our setting differs in that the rollout consists of **hidden-state sequences rather than token sequences**. Under this formulation:
>
> - For positive-reward rollouts $A>0$: Interpreting $\pi$ as the joint probability of the target item is intuitive and aligns directly with the recommendation objective.
> - For negative-reward rollouts $A<0$: Directly optimizing the probability of the target item may seem counterintuitive.  **In fact, we have experimented with different alternatives, e.g., optimizing the probability of the top negative item for $A<0$, or always optimizing the top-1 item regardless of the sign.**
>
> These variants, however, led to **overfitting or weaker reward alignment**. Details are provided in **Section 3.4.3** of the revised version.
>
> While a more formal theoretical justification would indeed be valuable, we view this as a promising direction for future exploration.

---

> ### Author Response · Authors · 2025-11-24
> **Official Response (Part 2)**
>
> > **Q2: Comparison with LatentR3**
>
> A2: LatentR3 proposes a reinforced latent reasoning framework for **LLM-based recommendation**, but LARES is designed for **pure recommendation models**. And we are concurrent works. Generally, it differs from LARES in three key aspects:
>
> 1. **Latent Reasoning Paradigm**:
>    - LatentR3 follows the paradigm of ReaRec, enhanced with an additional attention head.
>
>    - In contrast, LARES uses a recurrent architecture that updates all hidden states at each reasoning step.
>
> 2. **Training efficiency**:
>    - LatentR3 relies on LLMs (~1.5B parameters), leading to significantly longer training times and higher inference latency.
>
>    - Their experiments were conducted on much **smaller datasets** (around 5,000 items and 100,000 interactions) compared to ours.
>
> 3. **Reinforcement Learning Design**:
>    - **Reward:** LatentR3 uses perplexity as a reward, which is not well-suited for item matching, as shown in our experiments. We use **ranking metrics** as rewards, better aligning with the recommendation objective.
>    - **Supervision:** LatentR3 only computes loss for the response part, without supervision for the intermediate reasoning process. In contrast, **LARES calculates loss at all steps of reasoning to ensure coherence throughout the process.**
>    - **Rollouts:** LatentR3 generates diverse rollouts by injecting noise. LARES explores diverse reasoning trajectories through varying initial state initializations and reasoning depths, **offering a broader and more meaningful exploration compared to noise-based perturbations**, which can introduce irrelevant or meaningless trajectories.
>
> We evaluate the performance of LatentR3 and LARES on the Instrument and Scientific datasets. For a fair comparison, **we reproduce LatentR3 using the official public implementation, adapting it to SASRec under the same experimental settings as those in our paper**. We also fine-tune its loss coefficients following the guidelines in the LatentR3 paper.
>
> In addition, we experiment with two different RL reward signals for LatentR3: perplexity (PPL) and Recall@10. As shown in Table 2, LARES consistently outperforms LatentR3 on both datasets, demonstrating its effectiveness.
>
> Furthermore, we observe that using Recall@10 as the reward yields better results than perplexity. This is likely because perplexity reflects only the absolute likelihood of the target item and fails to capture relative ranking information, making it less aligned with the ranking-based nature of recommendation tasks.
>
> **Table 2. Performance Comparison of LatentR3 and LARES**
>
> | Dataset    | Metric    | SASRec | LatentR3 | +GRPO(PPL) | +GRPO(Recall) | LARES  | +GRPO       |
> | - | - | - | - | - | - | - | - |
> | Instrument | Recall@5  | 0.0346 | 0.0360   | 0.0362     | 0.0368        | 0.0396 | **0.0411*** |
> |            | R@10 | 0.0536 | 0.0554   | 0.0568     | 0.0573        | 0.0624 | **0.0636*** |
> |            | R@20 | 0.0798 | 0.0831   | 0.0839     | 0.0846        | 0.0925 | **0.0934*** |
> |            | N@5    | 0.0216 | 0.0233   | 0.0235     | 0.0239        | 0.0252 | **0.0263*** |
> |            | N@10   | 0.0277 | 0.0295   | 0.0301     | 0.0305        | 0.0326 | **0.0336*** |
> |            | N@20   | 0.0343 | 0.0364   | 0.0369     | 0.0370        | 0.0401 | **0.0410*** |
> | Scientific | R@5  | 0.0248 | 0.0255   | 0.0255     | 0.0263        | 0.0290 | **0.0297*** |
> |            | R@10 | 0.0385 | 0.0401   | 0.0403     | 0.0408        | 0.0456 | **0.0464*** |
> |            | R@20 | 0.0583 | 0.0600   | 0.0605     | 0.0610        | 0.0682 | **0.0705*** |
> |            | N@5    | 0.0150 | 0.0162   | 0.0163     | 0.0167        | 0.0180 | **0.0191*** |
> |            | N@10   | 0.0194 | 0.0209   | 0.0210     | 0.0214        | 0.0234 | **0.0245*** |
> |            | N@20   | 0.0244 | 0.0259   | 0.0261     | 0.0265        | 0.0291 | **0.0305*** |
>
> We also reproduce **LatentR3 based on Qwen2.5-1.5B-Instruct** to align with the original paper's settings. Due to time constraints, we only ran it on the Scientific. The performance, parameter count, and inference time are reported in Table 3, with LatentR3's inference time calculated using 4 NVIDIA RTX 3090 GPUs. The results show that LARES outperforms LatentR3, **achieving better performance with significantly fewer parameters and much faster inference.** This highlights LARES' superiority in both performance and efficiency compared to LLM-based methods.
>
> **Table 3. Performance comparison of LatentR3 and LARES**
>
> |               | #Param | #FLOPs | Inference Time | R@5   | R@10  | N@5     | N@10    |
> |  -| - | - | - | - | - | - | - |
> | LatentR3+SFT  | 1.5B   | 1.54T  | 4.6h           | 0.0249     | 0.0369     | 0.0158     | 0.0197     |
> | LatentR3+GRPO | 1.5B   | 1.54T  | 4.6h           | 0.0261     | 0.0388     | 0.0165     | 0.0207     |
> | LARES         | 1.9M   | 8M     | 1.51s          | **0.0297** | **0.0464** | **0.0191** | **0.0245** |

---

> ### Author Response · Authors · 2025-11-24
> **Official Response (Part 3)**
>
> > **Q3: Supplement computational overhead experiments**
>
> A3: We report the computational overhead in **Table 4 of Appendix A.5**, including parameters, FLOPs, GPU memory, and inference time on Instrument under the same batch size and environment settings. It can observed that:
>
> - LARES achieves **similar inference latency and memory usage as SASRec** at matched FLOPs.
> - ReaRec incurs additional inference latency and much **higher GPU memory consumption** due to its reasoning and KV-cache mechanisms.
>
> > **Q4: Ablation of reward functions and data selection strategies**
>
> A4: We conduct ablation studies on reward functions and data selection strategies using the Instrument and Scientific datasets. Specifically, the results for reward function variants are reported in **Appendix A.6**.
>
> We experiment with multiple reward signals, including Recall@{5, 10, 20} and NDCG@{5, 10, 20}. Our findings show that rewards with \( K > 10 \) consistently underperform those with \( K = 5 \) or \( K = 10 \), indicating that **the granularity of the reward (i.e., the choice of ranking cutoff) can influence the effectiveness of RL training.**
>
> Table 4 presents the results of different data selection strategies. Incorporating the Top-100 item filtering **reduces training time by more than 40% on both datasets** while also yielding improved final performance. This improvement likely stems from the sparsity of rewards in the original datasets, which can hinder effective RL training. These results strongly demonstrate the effectiveness of our proposed data selection strategy.
>
> **Table 4. Efficiency and Performance of Data Selection Strategies**
>
> |                        |         # Train Data |       Sec/Epoch | Training Time (h) |       R@10 |       N@10 |
> | :--------------------- | -------------------: | --------------: | ----------------: | ---------: | ---------: |
> | Instrument             |              339,519 |             187 |              2.70 |     0.0631 |     0.0332 |
> | +data select (top-100) | **144,062 (-57.6%)** | **80 (-57.2%)** | **1.62 (-40.0%)** | **0.0636** | **0.0336** |
> | Scientific             |              259,992 |             102 |              1.72 |     0.0454 |     0.0238 |
> | +data select (top-100) | **120,775 (-53.6%)** | **47 (-53.9%)** | **0.92 (-46.5%)** | **0.0464** | **0.0245** |

---

> ### Author Response · Authors · 2025-11-28
> **Looking forward to your reply!**
>
> Dear Reviewer Z3f7,
>
> Thank you for your time and thoughtful feedback on our paper. We sincerely appreciate the effort you’ve taken to review our work. We have carefully addressed all of your concerns in our response and hope it meet your expectations.
>
> If you have any further suggestions or questions, please don’t hesitate to let us know.

---

### Official Review · Reviewer_eZEV · 2025-11-07

**Soundness:** 2
**Presentation:** 3
**Contribution:** 2
**Rating:** 4
**Confidence:** 4

**Summary:**

This paper proposes a sequence recommendation framework called LARES, which significantly improves the model's expressive and recommendation performance without increasing parameters by performing multi-step recurrent reasoning in the latent space. The model employs a two-stage training approach: self-supervised pre-training to learn latent reasoning patterns, followed by reinforcement learning post-training to further optimize reasoning capabilities through reward signals. Experimental results show that LARES significantly outperforms existing methods on multiple real-world datasets, validating its effectiveness in deep reasoning and recommendation tasks.

**Strengths:**

This paper is well-written, organized, and clear, with excellent illustrations. The experimental section provides detailed discussion and analysis, and the experimental improvements are significant compared to the baseline. On four real-world Amazon datasets, LARES significantly outperforms mainstream methods in metrics such as Recall@K and NDCG@K (improvements of 10%–20%), and the contribution of each module is verified through ablation experiments.

**Weaknesses:**

This paper claims to focus on reasoning for recommendation, but it lacks discussion and comparison with related works [1,2]. Compared to PRL, its main addition is an RL module, making the contribution relatively incremental. Moreover, although the paper claims to perform reasoning, in essence, it only passes the hidden representations through network blocks multiple times, which can hardly be considered genuine reasoning. I understand that the authors are trying to draw inspiration from the latent reasoning setup in Latent CoT, but unlike Latent CoT, which incorporates semantic supervision for chain-of-thought reasoning, this work lacks such semantic guidance signals. Therefore, the reasoning claim seems somewhat overstated.


[1] Rec-R1: Bridging Generative Large Language Models and User-Centric Recommendation Systems via Reinforcement Learning
[2] R1-Ranker: Teaching LLM Rankers to Reason

**Questions:**

See weaknesses.

---

> ### Author Response · Authors · 2025-11-24
>
> Thank you for your thorough evaluation. We have addressed the points you raised as follows.
>
> > **Q1: Discussion with related works Rec-R1 and R1-Ranker**
>
> A1: **Rec-R1 and R1-Ranker both aim to improve LLM-based recommendation via RL, which is a different research topic with ours.** Rec-R1 proposes an RL framework that optimizes LLMs with feedback from recommendation models. R1-Ranker similarly applies RL to improve LLM reasoning for ranking, including DRanker which generates ranking results in one step and IRanker which performs ranking by iterative elimination of the least relevant candidate.
>
> The key differences between LARES and Rec-R1 / R1-Ranker are as follows:
>
> - **Traning and inference efficiency**
>   - LARES explores latent reasoning with lightweight recommendation models (**~2M parameters**), making it practical for industrial-scale deployment.
>   - Rec-R1 and R1-Ranker rely on finetuning LLMs (**~3B parameters**), leading to substantially higher computational cost.
> - **Task Settings**
>   - Rec-R1 targets a **product search** scenario where the LLM generates queries and depends on an external retriever (e.g., BM25).
>   - R1-Ranker operates on a fixed, pre-constructed candidate set and **cannot scale to full item pools** because of limited context window.
>
> > **Q2: Discussion with ReaRec**
>
> A2: We have discussed it in the Introduction (lines 62-66). Here, we’d like to highlight two key differences between our work and ReaRec:
>
> - We propose a fundamentally **different latent reasoning paradigm based on a recurrent architecture**. ReaRec models latent reasoning in an autoregressive manner, refining only the last token at each step. In contrast, our approach perform latent reasoning by iteratively reusing selected blocks with backbone model, refining all token representations at every step to fully unleash computation capacity.
> - We explore **reinforcement learning** to enhance latent reasoning. Given the sparse nature of recommendation data, **obtaining high-quality annotated reasoning data is challenging** for supervised training. RL offers a promising direction by using rewards to guide self-improvement of the model's reasoning ability.
>
> > **Q3: Clarification of latent reasoning in recommendation**
>
> A3: We appreciate the opportunity to clarify our use of the term *reasoning*. We follow the definition of latent reasoning in work [1]. We think *generalized reasoning* is an **iterative self-refinement process**.
>
> - Both explicit chain-of-thought (CoT) and latent reasoning follow an iterative refinement process.
> - CoT exposes intermediate steps as natural-language tokens, whereas latent reasoning performs analogous iterative updates in hidden state space. **In essence, CoT is also a multi-step forward process**.
> - In our framework, the model refines its hidden behavioral representations over multiple steps, progressively moving predictions closer to the target. **This demonstrates that the model is not making a one-shot prediction but is actively improving its internal representation.** We therefore refer to our approach as latent reasoning–enhanced recommendation.
>
> To support this, we also provide a quantitative analysis evaluating outputs at different steps within the same forward pass. As shown in Table 1, performance consistently improves from step 1 to step 4, confirming the model’s iterative refinement behavior.
>
> **Table 1. Performance of different steps on Instrument**
>
> | Steps | Recall@10     | Recall@20     | NDCG@10       | NDCG@20       |
> | ----- | ------------- | ------------- | ------------- | ------------- |
> | 1     | 0.0595        | 0.0887        | 0.0312        | 0.0385        |
> | 2     | 0.0603        | 0.0891        | 0.0319        | 0.0391        |
> | 3     | 0.062         |0.0920          | 0.0330         | 0.0404|
> | 4     | **0.0636**    | **0.0934**    | **0.0336**    | **0.0410**    |
>
> [1] Scaling up Test-Time Compute with Latent Reasoning: A Recurrent Depth Approach. arXiv 2025.

---

> ### Author Response · Authors · 2025-11-28
> **Looking forward to your reply!**
>
> Dear Reviewer eZEV,
>
> Thank you for your time and thoughtful feedback on our paper. We sincerely appreciate the effort you’ve taken to review our work. We have carefully addressed all of your concerns in our response and hope it meet your expectations.
>
> If you have any further suggestions or questions, please don’t hesitate to let us know.

---

### Meta-Review · Area_Chair_dxFN · 2025-12-29

**Summary:**

1.Lack of discussions and comparisons with related works [1,2], proposed by Reviewer eZEV.

2.2.Overclaim of reasoning due to lack of semantic guidance signals, proposed by Reviewer eZEV and Reviewer K6R5.

3.Inadequate comparison with baselines, such as LatentR3, proposed by Reviewer Z3f7.

4.Lack of results on more complex, large-scale benchmarks such as Tmall or Yelp, proposed by Reviewer 1a8J.

5.Lack of comparisons of training efficiency against strong non-RL alternatives, proposed by Reviewer 1a8J.

**Reviewer Concerns:**

1.Lack of discussions and comparisons with related works [1,2], proposed by Reviewer eZEV: during the rebuttal phase, the authors claim that Rec-R1 and R1-Ranker studies different topics from this paper. However, the authors does not compare. Thus I think this point is still outstanding.

2.Overclaim of reasoning due to lack of semantic guidance signals, proposed by Reviewer eZEV and Reviewer K6R5: during the rebuttal phase, the authors claim that latent reasoning is iterative self-refinement process, which is a form of generalized reasoning. However, it is hard to define reasoning in recommendation, and the paper does not visualize the reasoning process. I recommend the authors change another concept, such as iterative self-refinement process. Thus I think this point is still outstanding.

3.Inadequate comparison with baselines, such as LatentR3, proposed by Reviewer Z3f7: during the rebuttal phase, the authors discuss the difference between LatentR3 and LARES. Experiments show that LARES outperforms LatentR3. Thus I think this point is addressed.

4.Lack of results on more complex, large-scale benchmarks such as Tmall or Yelp, proposed by Reviewer 1a8J: during the rebuttal phase, the authors add results on two new larger datasets, ml-1m and yelp2018. Thus I think this point is addressed.

5.Lack of comparisons of training efficiency against strong non-RL alternatives, proposed by Reviewer 1a8J:  during the rebuttal phase, the authors do not reply to this point. Thus I think this point is still outstanding.

**Reviewer Scores:**

Reviewer eZEV would keep his or her score as 4 if he or she has been able to participate fully in the discussion.

Reviewer Z3f7 would keep his or her score as 6 if he or she has been able to participate fully in the discussion.

Reviewer 1a8J would keep his or her score as 6 if he or she has been able to participate fully in the discussion.

Reviewer K6R5 would keep his or her score as 4 if he or she has been able to participate fully in the discussion.

---

### Decision · Program_Chairs · 2026-01-26

Reject